# STEMMUS-UEB v1.0.0: Integrated Modelling of Snowpack and Soil Water and Energy Transfer with Three Complexity Levels of Soil Physical Process

5    Lianyu Yu[1], Yijian Zeng[1], Zhongbo Su[1,2]

[1] Faculty of Geo-information Science and Earth Observation (ITC), University of Twente, Enschede, The Netherlands

[2] Key Laboratory of Subsurface Hydrology and Ecological Effect in Arid Region of Ministry of Education, School of Water and Environment, Chang'an University, Xi'an, China

10    *Correspondence to*: Yijian Zeng (y.zeng@utwente.nl); Zhongbo Su (z.su@utwente.nl)

**Abstract**

A snowpack has a profound effect on the hydrology and surface energy conditions of an area through its effects on surface albedo, roughness, and its insulating property. The modelling of a snowpack, soil water dynamics, and the coupling of the snowpack and underlying soil layer has been widely reported. However, the coupled liquid-vapor-air flow mechanisms considering the snowpack effect have not been investigated in detail. In this study, we incorporated the snowpack effect (Utah Energy Balance model, UEB) into a common modeling framework (Simultaneous Transfer of Energy, Mass, and Momentum in Unsaturated Soils with Freeze-Thaw, STEMMUS-FT), i.e., STEMMUS-UEB. It considers soil water and energy transfer physics with three complexity levels (from the basic coupled, to advanced coupled water and heat transfer, and further to the explicit consideration of airflow, termed BCD, ACD, and ACD-air, respectively). We then utilized in-situ observations and numerical experiments to investigate the effect of snowpack on soil moisture and heat transfer with the above-mentioned model complexities. Results indicated that the proposed model with snowpack can reproduce the abrupt increase of surface albedo after precipitation events while this was not the case for the model without snowpack. The BCD model tended to overestimate the land surface latent heat flux (LE). Such overestimations were largely reduced by ACD and ACD-air models. Compared with the simulations considering snowpack, there is less LE from no-snow simulations due to the neglect of snow sublimation. The enhancement of LE was found after winter precipitation events, which is sourced from the surface ice sublimation, snow sublimation, and increased surface soil moisture. The relative role of the mentioned three sources depends on the timing and magnitude of precipitation and the pre-precipitation soil hydrothermal regimes. The simple BCD model cannot provide a realistic partition of mass transfer flux. The ACD model, with its physical consideration of vapor flow, thermal effect on water flow, and snowpack, can identify the relative contributions of different components (e.g., thermal or isothermal liquid and vapor flow) to the total mass transfer fluxes. With the ACD-air model, the relative contribution of each component (mainly the isothermal liquid and vapor flows) to the mass transfer was significantly altered during the soil

thawing period. It was found that the snowpack affects not only the soil surface moisture conditions (surface ice and soil water content in the liquid phase) and energy-related states (albedo, LE) but also the transfer patterns of subsurface soil liquid and vapor flow.

## 1. Introduction

In cold regions, the snowpack has a profound effect on hydrology and surface energy through its change of surface albedo, roughness and insulating property (Boone and Etchevers, 2001; Zhang, 2005). Different than rainfall, the melted snowfall enters the soil with a significant lag in time, and a large and sudden outflow or runoff may be produced because of the snowmelt effect. The heat insulating property of snow cover also provides a buffer layer to reduce the magnitude of the underlying subsurface temperature variations and thus markedly affect the thickness of the active layer in cold regions. The effect of snow cover on the subsurface soils has been studied and reviewed (e.g., Zhang, 2005; Hrbáček et al., 2016). For instance, snow cover can act as an insulator between atmosphere and soil with its low thermal conductivity (Zhang, 2005; Hrbáček et al., 2016). The snowmelt functions as the energy sink by the absorption of heat due to phase change (Zhang, 2005). Yi et al. (2015) investigated the seasonal snow cover effect on the soil freezing/thawing process and its related carbon implications. Such studies mainly focus on the thermal effect of snowpack on the frozen soils. However, the effect of snowpack on the soil water and vapor transfer process is rarely reported (Hagedorn et al., 2007; Iwata et al., 2010; Domine et al., 2019).

Great amounts of modeling efforts have been made to better reproduce the snowpack characteristic and its effects. Initially, snowpack dynamics were expressed as a simple function of temperature. Nevertheless, these empirical relations have limited applications in complex climate conditions (Pimentel et al., 2015). Many physically-based models for the mass and energy balance in the snowpack have been developed for their coupling with hydrological models or atmospheric models. Boone and Etchevers (2001) divided these snow models into three main categories: i) simple force-restore schemes with the snow modeled as the composite snow-soil layer (Pitman et al., 1991; Douville et al., 1995; Yang et al., 1997) or a single explicit snow layer (Verseghy, 1991; Tarboton and Luce, 1996; Slater et al., 1998; Sud and Mocko, 1999; Dutra et al., 2010); ii) detailed internal-snow-process schemes with multiple snow layers of fine vertical resolution (Jordan, 1991; Lehning et al., 1999; Vionnet et al., 2012; Leroux and Pomeroy, 2017); iii) intermediate-complexity schemes with physics from the detailed schemes but with a limited amount of layers, which are intended for coupling with atmospheric models (e.g., Sun et al., 1999; Boone and Etchevers, 2001). The intercomparison results of the abovementioned snow models at an alpine site indicated that all three types of schemes are capable of representing the basic features of the snow cover over the 2-year period but behaved differently on shorter timescales. Furthermore, Snow Model Intercomparison Project (SnowMIP) at two mountainous alpine sites revealed that the albedo parameterization was the major factor influencing the simulation of net shortwave radiation. Though this parameterization is independent of model complexity (Etchevers et al., 2004) it directly affects the snow simulation. SnowMIP2 evaluated thirty-three snowpack models across a wide range of hydrometeorological and forest canopy conditions. It identified the shortcomings of different snow models and highlighted the necessity of studying the separate contribution of individual components to the mass and energy balance of snowpack (Rutter et al., 2009). With the majority of research focuses on the intercomparison of the snowpack models with various physical complexities, little attention has been paid to

the treatment of the underlying soil physical processes (see the brief overview of current soil-snow modelling efforts in Table 1).

In the current soil-snow modeling research, soil water and heat transfer are usually not fully coupled and moreover the vapor flow and airflow are absent (Koren et al., 1999; Niu et al., 2011; Swenson et al., 2012). This may lead to the unrealistic interpretation of the underlying soil physical processes and the snowpack energy budgets (Su et al., 2013; Wang et al., 2017). Researchers have emphasized the need to consider the coupled soil water and heat transfer mechanisms (Scanlon and Milly, 1994; Bittelli et al., 2008; Zeng et al., 2009a; Zeng et al., 2009b; Yu et al., 2018). In consequence, dedicated efforts have been made to implement it in the recent updated models (e.g., Painter et al., 2016; Wang et al., 2017; Cuntz and Haverd 2018). On the other hand, the role of the airflow has been reported important in many relevant studies, including retarding soil water infiltration (Touma and Vauclin, 1986; Prunty and Bell, 2007), enhancing surface evaporation after precipitation (Zeng et al., 2011a, b), enlarging the temperature difference between the upper and lower part of a permafrost talus slope (Wicky and Hauck, 2017), interacting with soil ice and vapor components and enhancing the vapor transfer in frozen soils (Yu et al., 2018; Yu et al., 2020). However, to our knowledge, few soil-snow models have taken into account the soil dry air transfer processes and moreover the multi-parameterization of the soil physical processes (from the basic coupled to the advanced coupled water and heat transfer processes and, then, to the explicit consideration of airflow), resulting in the lack of understanding on how and to what extent the complex soil physics affect the model interpretation of the snowpack effects.

In this paper, one of the widely used snowpack models (Utah energy balance snowpack model, UEB, Tarboton and Luce, 1996) was incorporated into a common soil modeling framework (Simultaneous Transfer of Energy, Mass and Momentum in Unsaturated Soils with Freeze-Thaw, STEMMUS-FT, Zeng et al., 2011a, b; Zeng and Su, 2013; Yu et al., 2018). The new model is named STEMMUS-UEB and is configured with various levels of model complexity in terms of mass and energy transport physics. We utilized in situ observations and numerical experiments with STEMMUS-UEB to investigate the effect of snowpack on the underlying soil mass and energy transfer with different complexities of soil models. The description of the coupled soil-snow modeling framework STEMMUS-UEB and the model setup for this study are presented in Section 2. Section 3 verified the proposed model and identified the effect of snowpack on soil liquid/vapor fluxes. The uncertainties and limitations of this study and the applicability of the proposed model are discussed in Section 4.

## 2 Description of Coupled Soil-Snow Modelling Framework and Model Setup

This section first presents the coupling procedure of STEMMUS-FT and UEB model, followed by the detailed description of the two models and their successful applications. Then the used model configurations and two tested experimental sites in the Tibetan Plateau were elaborated. Maqu case is for investigating the effect of snowpack on the underlying soil hydrothermal regimes. Yakou case is for demonstrating the validity

of the developed STEMMUS-UEB model in reproducing the snowpack dynamics (results were presented in Appendix B). In addition, the relationship between the snow cover properties and albedo was presented in Appendix B.4, which confirmed the validity of using the albedo to identify the presence of snowpack and its lasting time.

**2.1 Coupling procedure**

The coupled process between the snowpack model (UEB) and the soil water model (STEMMUS-FT) was illustrated in Figure 1. The sequential coupling is employed to couple the soil model with the current snowpack model. The role of the snowpack is explicitly considered by altering the water and heat flow of the underlying soil. The snowpack model takes the atmospheric forcing as the input (precipitation, air temperature, wind speed and direction, relative humidity, shortwave and longwave radiation) and solves the snowpack energy and mass balance (Eq. A.8 & A.9, Subroutines: ALBEDO, PARTSNOW, PREDICORR), provides the melt water flux and heat flux as the surface boundary conditions for the soil model STEMMUS-FT (Subroutines: h_sub and Enrgy_sub for ACD models; Diff_Moisture_Heat for BCD model). The soil-snow coupling variables are the snowmelt water flux $M_r$, the convective heat flux due to snowmelt water $Q_m$ and the heat conduction flux $Q_g$. STEMMUS-FT then solves the energy and mass balance equations of soil layers in one time step. To highlight the effect of the snowpack on the soil water and vapor transfer process, we constrained the soil surface energy boundary as the Dirichlet type condition (take the specific soil temperature as the surface boundary condition). Surface soil temperature was derived from the soil profile measurements and was not permitted to be higher than zero when there is snowpack. In such way, the reliability of the soil surface energy boundary condition is maintained and the snow thermal effect is implicitly considered. The snowmelt water flux, in addition to the rainfall, was added to the topsoil boundary for solving soil water transfer. To ensure numerical convergence, the adapted time step strategy was used. Half-hourly meteorological forcing measurements were linearly interpolated to the running timesteps (Subroutine Forcing_PARM). The precipitation rate (validated at 3-hour time intervals) was regarded uniformly within the 3-hour duration (refer to Table S1 for detail). The general description of the primary subroutines in STEMMUS-UEB was presented in Table 2. It includes the main functions, input/output, and their connections with other subroutines (linked with Table S1 and S2 for the description of model input parameters and outputs for this study, see the detailed description in Tarboton and Luce, 1996; Zeng and Su, 2013; Yu et al., 2018).

**2.2 Soil mass and heat transfer module**

The detailed physically based two-phase flow soil model (STEMMUS) was first developed to investigate the underlying physics of soil water, vapor, and dry air transfer mechanisms and their interaction with the atmosphere (Zeng et al., 2011a, b; Zeng and Su, 2013). It is achieved by simultaneously solving the balance equations of soil mass, energy, and dry air in a fully coupled way. The mediation effect of vegetation on such interaction was latterly incorporated via the root water uptake sub-module (Yu et al., 2016) and furthermore

by coupling with the detailed soil and vegetation biogeochemical process (Wang et al., 2021; Yu et al., 2020a). It facilitates our understanding of the hydrothermal dynamics of respective components in the frozen soil medium (i.e., soil liquid water, water vapor, dry air, and ice) by implementing the freeze-thaw process (hereafter STEMMUS-FT, for applications in cold regions, Yu et al., 2018; Yu et al., 2020c).

The frozen soil physics considered in STEMMUS-FT include three parts: i) the ice blocking effect on soil hydraulic conductivities (see Supplement Sect. 2.2.2); ii) the inclusion of ice effect in the calculation of soil thermal capacity/conductivity (see Supplement Sect. 2.2.8); iii) the exchange of latent heat flux during phase change periods. With the aid of Clausius Clapeyron relation, which characterizes the phase transition between liquid and solid phase in the thermal equilibrium system, the soil water characteristic curve (e.g., van Genuchten, 1980) is then extended to consider the freezing temperature dependence, i.e., soil freezing characteristic curve (Hansson et al., 2004; Dall'Amico et al. 2011). The fraction of soil liquid/solid water at a given temperature was then calculated prognostically with the soil freezing characteristic curve. Soil hydraulic parameters were further used in the Mualem (1976) model to compute the soil hydraulic conductivity. The ice effect is considered by reducing the soil saturated hydraulic conductivity as the function of ice content (Yu et al., 2018).

In response to minimize the potential model-comparison uncertainties from various model structures and to figure out which process matters, three levels of complexity of mass and heat transfer physics are made available in the current STEMMUS-FT modelling framework (Yu et al., 2020c). First, the 1-D Richards equation and heat conduction were deployed in STEMMUS-FT to describe the isothermal water flow and heat flow (termed BCD). The BCD model considers the interaction of soil water and heat transfer implicitly via the parameterization of heat capacity, thermal conductivity, and the water phase change effect. The water flow is fully affected by soil temperature regimes in the advanced coupled water and heat transfer model (termed ACD model). The movement of water vapor, as the primary linkage between soil water and heat flow, is explicitly characterized. STEMMUS-FT further enables the simulation of temporal dynamics of three water phases (liquid, vapor, and ice), together with the soil dry air component (termed ACD-air model). The governing equations of liquid water flow, vapor flow, airflow, and heat flow were listed in Appendix A.1 (see the more detailed model description in Zeng et al., 2011a, b; Zeng and Su, 2013; Yu et al., 2018; Yu et al., 2020c).

**2.3 Snowpack module UEB**

The Utah energy balance (UEB) snowpack model (Tarboton and Luce, 1996) is a single-layer physically-based snow accumulation and melt model. Two precipitation types, i.e., rainfall and snowfall, are discriminated by its dependence on air temperature. The snowpack is characterized using two primary state variables, snow water equivalent *SWE* and the internal energy *U*. Snowpack temperature is expressed diagnostically as the function of *SWE* and *U*, together with the states of the snowpack (i.e., solid, solid and liquid mixture, and liquid). Given the insulation effect of the snowpack, snow surface temperature differs

from the snowpack bulk temperature, which is mathematically considered using the equilibrium method (i.e., balances energy fluxes at the snow surface). The age of the snow surface, as the auxiliary state variable, is utilized to calculate the snow albedo (see Appendix A.3). When the snowpack is shallow, the albedo is the weighting function of the snow albedo and the bare ground albedo. The solar radiation penetration in the shallow snowpack is exponentially attenuated and expressed in the weighting factor. The melt outflow is calculated using Darcy's law with the liquid fraction as inputs. The conservation of mass and energy forms the physical basis of UEB (Tarboton and Luce, 1996, as presented in Appendix A.2).

UEB is recognized as one simple yet physically-based snowmelt model. It captures the snow process well (e.g., diurnal variation of meltwater outflow rate, snow accumulation, and ablation, see a general overview of UEB model development and applications in Table S3). It requires little effort in parameter calibration and can be easily transferable and applicable to various locations (e.g., Gardiner et al., 1998; Schulz and de Jong, 2004; Watson et al., 2006; Sultana et al., 2014; Pimentel et al., 2015; Gichamo and Tarboton, 2019), especially for data scarce regions as for example Tibetan Plateau. We thus selected the original parsimonious UEB (Tarboton and Luce, 1996) as the snow module to be coupled with the soil module (STEMMUS-FT).

**2.4 Configurations of numerical experiments**

On the basis of the aforementioned STEMMUS-UEB coupling framework, the various complexities of vadose zone physics were further implemented as three alternative model versions. First, the soil ice effect on soil hydraulic and thermal properties, and the heat flow due to the water phase change were taken into account, while the water and heat transfer is not coupled in STEMMUS-FT and termed the BCD model. Second, the STEMMUS-FT with the fully coupled water and heat transfer physics (i.e., water vapor flow and thermal effect on water flow) was applied and termed the ACD model. Lastly, on top of the ACD model, the air pressure was independently considered as a state variable (therefore, the airflow) and termed the ACD-air model. With the abovementioned model versions (STEMMUS-FT_Snow) and taking into account the no-snow scenarios (STEMMUS-FT_No-Snow), Table 3 lists the configurations of all six designed numerical experiments. The model parameters used for all simulations for the tested experimental site are listed in Table S2.

**2.5 Description of the Tested Experimental sites**

Maqu station, equipped with a catchment scale soil moisture and soil temperature (SMST) monitoring network and micro-meteorological observing system, is situated on the north-eastern edge of the Tibetan Plateau (Su et al., 2011; Dente et al., 2012; Zeng et al., 2016). According to the updated Köppen-Geiger climate Classification System, it can be characterized as a cold climate with dry winter and warm summer. The annual mean precipitation is about 620 mm and the annual average evaporation is about 1353.4 mm. Precipitation in Maqu is uneven within the year with most of the precipitation events occurs from May to October and little precipitation/snowfall during the wintertime. The average annual air temperature is 1.2 ℃,

and the mean air temperatures of the coldest month (January) and the warmest month (July) are about -10.0 ℃ and 11.7 ℃, respectively. Alpine meadows (e.g., *Cyperaceae* and *Gramineae*), with a height varying from 5 cm to 15 cm throughout the growing season, are the dominant land cover in this region. This site is seasonally snow covered with the temporal snow in the non-growing season, which is due to the less and intermittent snowfall and the rapid snow melting and sublimation as the high air temperature and strong solar radiation in the daytime. The general soil types are sandy loam, silt loam and organic soil for the upper soil layers (Dente et al., 2012; Zheng et al., 2015; Zhao et al., 2018). The soil texture and hydraulic properties were listed in Table S2 and how it was used in STEMMUS-UEB is illustrated in Figure 1 and Table 2.

The Maqu SMST monitoring network spans an area of approximately 40 km×80 km with the elevation ranging from 3200 m to 4200 m a.s.l. (33°30′–34°15′N, 101°38′–102°45′E). SMST profiles are automatically measured by 5TM ECH$_2$O probes (METER Group, Inc., USA) installed at different soil depths, i.e., 5 cm, 10 cm, 20 cm, 40 cm, and 80 cm. The micro-meteorological observing system consists of a 20 m Planetary Boundary Layer (PBL) tower providing the meteorological measurements at five heights aboveground (i.e., wind speed and direction, air temperature and relative humidity), and an eddy-covariance system (EC150, Campbell Scientific, Inc., USA) equipped for measuring the turbulent sensible and latent heat fluxes and carbon fluxes. The equipment for four-component down and upwelling solar and thermal radiation (NR01-L, Campbell Scientific, Inc., USA), and liquid precipitation (T200B, Geonor, Inc., USA) are also deployed. The dataset from 1 December 2015 to 15 March 2016 was utilized in this study. An independent precipitation data (3-hour time interval) during the same testing period from an adjacent meteorological station was used as the mutual validation data.

Yakou super snow station (38°00′36″N, 100°14′24″E, 4145 m) is located in the upstream Heihe River basin, northeastern of Tibetan Plateau. It is a high-elevation snow covered site with the wet summers and dry winters. The dominant land type is Tundra with frozen ground below. There is a unique seasonal variation of snow depth with the maximum snow depth usually in the springtime (32 cm during the period 2014-2017). Loam is the main soil type with the silt loam near the surface and sandy soil for the deeper soil layers.

The integrated hydrometeorological, snow cover and frozen ground data was published and available from the Cold and Arid Regions Science Data Center at Lanzhou (Che et al., 2019; Li et al., 2019; Li, 2019). The meteorological data (air temperature, wind speed, precipitation, downward shortwave/longwave radiation, and relative humidity) was recorded by the automatic meteorological station (AMS). In situ measurements of snow cover properties (snow depth and snow water equivalent) were obtained using the state-of -the-art instruments (SR50A and GammaMONitor, Campbell Scientific, USA). Soil moisture profiled at 4, 10, 20, 40, 80, 120, and 160 cm soil depth was measured using ECH2O-5 probes (METER Group, Inc., USA). In addition to the seven soil depths, the surface soil temperature (0 cm) was also recorded using the Avalon AV-10T sensors (Avalon Scientific, Inc., USA). The eddy covariance system was equipped in the Yakou site for measuring land surface turbulent fluxes. The dataset from 1 September 2016 to 31 December 2016 was used to validate the model performance in mimicking the dynamics of snow water equivalent, soil hydrothermal

regimes and land surface evaporation. The calibrated soil hydraulic properties, snow cover properties were listed in the Supplement II Table S2.

## 3. Results: comparison of simulation results of surface variables with/without snowpack effect

### 3.1 Albedo

The time series of surface albedo, calculated as the ratio of upwelling shortwave radiation to the downwelling shortwave radiation and estimated using BCD, ACD and ACD-air models, was shown in Figure 2 together with precipitation. As the snowpack has a higher albedo than the underlying surface (e.g., soil, vegetation), compared to the observations, models without snow module presented a relatively flat variation of daily average surface albedo, and lacked the response to the winter precipitation events (Figure 2, Table 4). With the snow module, STEMMUS-UEB models can capture mostly the abrupt increase of surface albedo after winter precipitation events. The mismatches in terms of the magnitude or absence of increased albedo after precipitation events indicated that the model tended to underestimate the albedo dynamics. The shallow snowfall events might be not well captured by the model (see Sect. 4.1). Three model versions (BCD-Snow, ACD-Snow, and ACD-air-Snow) produced similar fluctuations regarding the presence of snow cover with slight differences in terms of the magnitude of albedo.

### 3.2 Soil Temperature and Moisture Dynamics

The observed spatial and temporal dynamics of soil temperature from five soil layers was used to verify the performance of different models (Fig. 3). The initial soil temperature state can be characterized as the warm bottom and cool surface soil layers (based on in-situ observations). The freezing front (indicated by the zero-degree isothermal line, ZDIL) developed downwards rapidly until the 70th day after December 1, 2015, reaching its maximum depth. Then the freezing front stabilized as the offset effect of latent heat release (termed as zero-curtain effect). Such influence can sustain until all the available water to that layer is frozen, at which point the latent heat effect is negligible compared to the heat conduction. At shallower layers, the atmospheric forcing dominates the fluctuation of thermal states. The isothermal lines (e.g., -2 °C) had a larger variation than that of ZDIL. At deeper soil layers, the temporal dynamics of isothermal lines were smoother than that of ZDIL, indicating that the effect of fluctuated atmospheric force on soil temperature was damped with the increase of soil depth. Compared to the observations, BCD-Snow model presented an earlier development of the freezing front and arrival of the maximum freezing depth (60th day after December 1, 2015). The deeper and more fluctuated freezing front indicates that a stronger control of atmospheric forcing on soil thermal states was produced by BCD-Snow model. The ACD models can well capture the propagation characteristic of the freezing front in terms of the variation magnitude and maximum freezing depth. There is no significant difference in soil thermal dynamics between the model with and without snow module, except at the surface soil layers (Table 4).

Figure 4 shows the spatial and temporal dynamics of observed and simulated soil water content in the liquid phase (SWCL). The SWCL of active layers depends to a large extent on the soil freezing/thawing status. Soil is relatively wet at soil layers of 10-60 cm for the starting period. Its temporal development was disrupted by the presence of soil ice and tended to increase wetness during the thawing period. A relatively dry zone ($\theta_L <$ $0.06 \ m^3 \ m^{-3}$) above the freezing front was found, indicating the nearly completely frozen soil during the stabilization stage. The initial wet zone of soil moisture was narrowed down and the rewetting zone tended to enlarge from BCD-Snow simulation due to its early freezing and thawing of soil (Fig. 4b). The position of the dry zone occurred earlier as the early reaching of the stabilization period by the BCD-Snow model (Fig. 3b). For the ACD models, the position and development of initial wet zone, rewetting zone and the dry zone is similar to that from the observations, indicating the soil moisture dynamics can be well captured by the ACD models. Compared to the STEMMUS-FT_Snow model, there was no observable difference in the SWCL dynamics at deeper soil layers from STEMMUS-FT_No-Snow simulations. The surface SWCL was found affected from STEMMUS-FT_Snow simulations (Table 4).

**3.3 Surface Latent Heat Flux**

Figure 5 shows the comparison of time series of observed and model simulated surface cumulative latent heat flux using three models with/without consideration of snow module. Considerable overestimation of latent heat flux was produced by the BCD-Snow model, with 121.79% more than observed. Such overestimations were largely reduced by ACD and ACD-air models. There is a slight underestimation of cumulative latent heat flux by ACD-Snow and ACD-air-Snow models, with -8.33% and -7.05%, respectively. Compared with STEMMUS-FT_Snow simulations, there is less latent heat flux produced by STEMMUS-FT_No-snow simulations. It is mainly due to the sublimation of snow cover, which cannot be simulated by the STEMMUS-FT_No-snow models. The difference in cumulative latent heat flux between STEMMUS-FT with and without snow module increases from BCD to ACD-air schemes, with the values of 2.02%, 7.69%, and 8.97% for BCD, ACD and ACD-air schemes, respectively.

**3.4 Liquid/vapor fluxes**

To further elaborate the effect of snowpack on LE, we presented the diurnal variations of LE and its components at two typical episodes with precipitation events (freezing and thawing period, respectively). The relative contribution of liquid and vapor flow to the total mass transfer after precipitation events was separately presented in Figure 8 & 9, i.e., the liquid water flux driven by temperature $q_{LT}$, matric potential $q_{Lh}$ and air pressure $q_{La}$, water vapor flux driven by temperature $q_{VT}$, matric potential $q_{Vh}$ and air pressure $q_{Va}$.

**1) LE**

Diurnal dynamics of the observed and simulated latent heat flux during the rapid freezing period with the occurrence of precipitation events, from 10th to 14th Days after Dec. 1. 2015, is shown as Fig. 6a, b &c. Compared to the observations, the diurnal variations of latent heat flux were captured by the proposed model

with various levels of complexities. Performance of BCD, ACD, and ACD-air models in simulating LE differed mainly regarding the magnitude and response to precipitation events. For the BCD-Snow model, the overestimation of LE was found at 10th and 11th day after December 1 due to relatively high surface soil moisture simulation (Fig. S1b). A certain amount of enhanced surface evaporation was produced shortly after precipitation, which is most probably due to the snow sublimation. Snow sublimation presents appear not intuitively matching with observations. The mismatch in the LE enhancement after precipitation events can be attributed to that the partition process of precipitation into various components (rainfall, snowfall, canopy interception) might not be well captured by the model. Such a response to the winter precipitation events was absent from the BCD-No-Snow simulations.

The overestimation of LE was reduced by ACD and ACD-air models (Fig. 6b & c). Compared to the ACD-Snow model simulations, ACD-No-snow model produced a stronger diurnal variation of LE after the precipitation and is more approaching the measured LE. Lower diurnal variation of LE for the ACD-Snow model can be ascribed to the lower surface SWCL (see Fig. S1d & g). For the ACD-Snow model, precipitation was partitioned into rainfall and snowfall, part of which was directly evaporated as sublimation. The sum of rainfall and the melting part of snowfall reached the soil surface as the incoming water flux, which is less than that for the ACD-No-snow model (took all the precipitation as the incoming water flux). There is no significant difference in the dynamics of LE between simulations by ACD models and ACD-air models.

During the thawing period, the diurnal variations of LE were well simulated by the models (Figure 7). There are some discrepancies regarding the peak values of LE. For the BCD-Snow model, overestimations were found in 100th, 101st, and 102nd day after December 1, 2015. The high LE values on 100th and 101st day are probably due to the high surface soil moisture by the thawing water (Fig. S2b). While on the 102nd day, it is due to the snow sublimation (Fig. 7a). The peak values were reproduced but shifted by BCD-No-Snow simulation, which occurred on 100th and at the end of 102nd, indicating the shift of surface soil moisture states (Fig. S2b).

For the ACD model, the difference in latent heat flux between snow and no-snow simulations was noticeable two days after precipitation. The larger values of LE from the ACD-No-snow model occurred earlier than that from the ACD-Snow model, as the earlier response of surface soil moisture to the precipitation event (Fig. S2). While compared to the observations, the enhancement of LE advanced from the ACD-Snow simulations (Fig. 7b). This enhanced evaporation can be attributed to the snow sublimation and increased surface soil moisture content. Similar lag behavior of precipitation-enhanced evaporation was produced by the ACD-air-Snow models (Figure 7c). There are mismatches in the time and magnitude of LE enhancement between ACD-Snow model simulations and observations (Fig. 7b). This discrepancy lies in the uncertainties of snowpack simulations, which can be attributed to either the inaccurate precipitation measurements (Barrere et al., 2017; Günther et al., 2019) or that the precipitation partition process is not well described by the model (Harder and Pomeroy, 2014; Ding et al., 2017).

**2) LE and decomposition of surface mass transfer**

During the freezing period, the soil water vapor, instead of liquid water flux, dominated the surface mass transfer process. Missing the description of the vapor diffusion process hindered the BCD models to realistically depict the decomposition of surface mass transfer dynamics (Fig. 8a &b).

There is a visible diurnal variation of thermal vapor flux $q_{VT}$ from the ACD model simulation (Fig. 8c &d). The isothermal vapor flux $q_{Vh}$ contributed to most of the mass transfer during the freezing period. It should be noted that the sum of water/vapor fluxes at 0.1cm soil layer cannot balance the surface evaporation, especially after the precipitation events (Fig. 8c). We assumed and attributed it to the surface ice sublimation process. Precipitation water was frozen on the soil surface, and only vapor fluxes are active in the topsoil layers. Sublimation of surface ice may contribute to the gaps between liquid/vapor fluxes and LE (Yu et al., 2018). As more precipitation water was frozen on the soil surface from the ACD-No-Snow model (Fig. 8d), the difference between the sum of water/vapor fluxes at the top 0.1cm soil layer and the surface evaporative water enlarged compared to ACD-Snow simulations. Thermal liquid water flux $q_{LT}$ appears negligible to the total mass flux during the whole simulation period. There is no significant difference recognized in the mass transfer between the ACD-air and ACD during the freezing period.

During the thawing period, a certain amount of upward liquid water flux was produced by the BCD model, supplying the water to the topsoil and evaporate into the atmosphere (Fig. 9a &b). Compared to the isothermal liquid flux $q_{Lh}$, the thermal liquid flux $q_{LT}$ was negligible to the total mass flux.

For the ACD model, the diurnal variation of thermal vapor flux $q_{VT}$ was enhanced after precipitation, producing a larger amount of upward/downward vapor flux during the night/daytime (e.g., Fig. 9c). As the surface soil is relatively dry, the isothermal vapor flux $q_{Vh}$ contributes nearly all the mass flux during the selected thawing period. Driven by the matric potential gradient, a large amount of isothermal water vapor flux $q_{Vh}$, accompanied by downward liquid water flux $q_{Lh}$, can be found after the nighttime precipitation event (Fig. 9c, d, e, f). These precipitation-induced isothermal liquid/vapor fluxes were lagged and less intense from the ACD-Snow model than that from the ACD-No-Snow model simulation (e.g., Fig. 9c vs. Fig. 9d). It is explained that the snowpack reduces the instant precipitation infiltration process and enables the snowmelt afterwards, which led to the lagged and weaker response of surface SWCL to the precipitation (Fig. S2). It breaks the balance between isothermal vapor flux and evaporative LE (around 103$^{rd}$ day after Dec. 1, 2015). Compared to the ACD-No-Snow model, such imbalance was enlarged for the ACD-Snow model during the thawing period (Fig. 9c &d).

Compared to the ACD-No-Snow simulations, the upward thermal vapor flux $q_{VT}$ was enhanced after precipitation for the ACD-air-No-Snow model (Fig. 9f). This enhanced upward vapor flux reduced the soil liquid water content at 0.1cm (Fig. S2f) and decreased the soil hydraulic conductivity and then the downward isothermal liquid/vapor flux ($q_{Lh}$, $q_{Vh}$). Other than that, there is no significant difference between the ACD-air model and the ACD model during the thawing period.

## 4. Discussion

### 4.1 Uncertainties in simulations of surface albedo and limitations

After a winter precipitation event, land surface albedo increases considerably (Fig. 2), indicating the presence
of the snowpack. However, such snowfall events were episodic with small magnitude (similar in Li et al.,
2017), which is difficult to be well captured. Such difficulties can be partially attributed to the inherent
uncertainties in precipitation measurements (both the precipitation amount and types). Due to the spatial
variability of precipitation, the accurate observation of winter precipitation is proved to be a challenge,
especially during windy winters (Barrere et al., 2017; Pan et al., 2017). It is necessary to have more
snowpack-relevant measurements (e.g., the high-resolution measurements of the spatiotemporal field of wind
speed, precipitation, and snowpack variations) to understand the dynamics of snowpack and its effect on
energy and water fluxes. Furthermore, the temporal resolution of precipitation measurements adopted in this
study is relatively coarse (3-hour). In the current precipitation partition parameterization, the amount of
snowfall was determined as a function of precipitation and air temperature thresholds. Given the coarse
temporal resolution of precipitation measurements, the model may produce a time shift of snowfall events or
even the mal-identification of snowfall. The simple relation between the air temperature and precipitation
types may be not suitable to this region, because air temperature is not the best indicator of precipitation
types, as argued by Ding et al. (2014). Other factors, i.e., relative humidity, surface elevation, and wet-bulb
temperature, are also very relevant and should be taken into account for the discrimination of precipitation
types. The other uncertainty lies in the representation of the snow process. For example, the wind-blow effect
and canopy snow interception, which have been recognized as important to the accurate simulation of
snowpack dynamics (Mahat and Tarboton, 2014), are not taken into account in detail. Last but not least, the
interpretation of surface albedo dynamics needs to be adapted to the specific site, especially regarding the
shallow snow situations (Ueno et al., 2007; Ueno et al., 2012; Ding et al., 2017; Wang et al., 2017). The
albedo of the underlying surface should also be properly accommodated to this Tibetan meadow system.
Regardless of the aforementioned uncertainties, our proposed model was capable to capture the surface
albedo variations with precipitation (Fig. 2) and can be seen as acceptable to analyze snow cover effects in
such a harsh environment.

### 4.2 Snow cover-induced evaporation enhancement

Different from the rainfall, precipitation water from snowfall enters the soil considerably lagged in time due
to the water storage by snow cover (You et al., 2019). With the snow module, precipitation was partitioned
into rainfall and snowfall. Part of the snowfall evaporated into the atmosphere as sublimation and the other
part together with the rainfall infiltrated into the underlying soil. It resulted in the delay of incoming water to
the soil with a less amount compared to that without consideration of the snow module. This amount of
incoming water increased the evaporation after precipitation (Fig. 6 & 7). The other source for the enhanced
evaporation flux after precipitation is snow sublimation, which is absent from the model without the snow

module. Sublimation occurs readily under certain weather conditions (e.g., with freezing temperatures, enough energy). It can be more active in regions with low relative humidity, low air pressure and dry winds. Such amount of sublimation has been reported important from the perspective of climate and hydrology (e.g., Strasser et al., 2008; Jambon-Puillet et al., 2018), especially at high altitude regions with the low air pressure. During the freezing period, the evaporation enhancement can be also sourced from the sublimation of surface ice. The amount of the ice sublimation appeared to decrease during the freezing period in the presence of a transient snowpack (e.g., Fig. 8c vs. 8d). This is consistent with the results of Hagedorn et al. (2007), who investigated the effect of snow cover on the mass balance of ground ice with an artificially continuous annual snow cover. According to their results, the snow cover enhanced the vapor transfer into the soil and thus reduced the long-term ice sublimation. The relative contribution of increased surface soil moisture, snow sublimation, and surface ice sublimation to the enhanced evaporation is dependent on the pre-precipitation soil moisture/temperature states, air temperature, and the time and magnitude of precipitation events. Under the conditions of the low pre-precipitation SWCL with the freezing soil temperature (e.g., Fig. 8e, 11th vs. 12th Days after 1 December), the precipitation falls on the surface as snowfall and rainfall (most freezes as ice). The sublimation from surface ice can contribute to most of the total mass transfer (e.g., Fig. 8e, 11th Days after 1 December). If the soil temperature rises above the freezing temperature, there will be no sublimation of surface ice, in terms of contributing to the enhanced evaporation (e.g., Fig. 9e, 102nd Days after 1 December).

**4.3 Snow cover impacts with different soil model complexities**

The model with different complexity of soil mass and energy transfer physics behaves differently in response to the winter precipitation events. During the freezing period, there is no significant difference in the BCD model simulated soil moisture with/without the snow module. The precipitation water freezes at the soil surface, which cannot be transferred downwards with the BCD model physics. The sublimation, from either the snow or the surface ice, contributes all to the precipitation-enhanced evaporation for the BCD model. As with vapor flow, the surface ice increases the soil moisture at lower layers via the downward isothermal vapor flux (Fig. 8). The surface ice sublimation and increased soil moisture-induced evaporation enhancement can be identified from the ACD model simulation. The role of airflow was negligible to the mass transfer during the freezing period.

When it comes to the thawing period, BCD model produced a certain amount of liquid water flow, contributing considerably to the mass transfer. The obvious fluctuation of SWCL was noticed due to the thawing water and precipitation event. The main source for the increased evaporation was interpreted as isothermal liquid water flow. While for the ACD model, the situation becomes more complex. Thawing surface ice and snowmelt water may coexist at the soil surface, resulting in different soil moisture response to precipitation events. The ice sublimation, snow sublimation, and increased soil moisture contribute to the evaporation enhancement after precipitation. When considering airflow, dry air interacts with soil ice,

liquid/vapor water in soil pores (Yu et al., 2018) and alters the soil moisture states. It thus considerably changes the relative contribution of each component to the mass transfer (Fig. 9).

## 5. Conclusions

With the aim to investigate the hydrothermal effect of the snowpack on the underlying soil system, we developed the integrated process-based soil-snow-atmosphere model, STEMMUS-UEB v1.0.0, which is based on the easily transferable and physically-based description of the snowpack process and the detailed interpretation of the soil physical process with various complexities. From STEMMUS-UEB simulations, snowpack affects not only the soil surface conditions (surface ice and SWCL) and energy-related states

(albedo, latent heat flux) but also the transfer patterns of subsurface soil liquid/vapor flow. STEMMUS-FT model can capture mostly the abrupt increase of surface albedo after winter precipitation events with consideration of the snow module. There is a significant overestimation of cumulative surface latent heat flux by the BCD model. ACD and ACD-air model produces a slight underestimation of cumulative LE compared to the observations. Without sublimation from snowpack, there is a less latent heat flux produced by

STEMMUS-FT_No-snow simulations compared with snow simulations. The presence of snowpack alters the partition process of precipitation and thus the surface SWCL. BCD models with/without snowpack produced similar surface SWCL during the freezing period while resulted in the abrupt increase of soil moisture in response to the precipitation during the thawing period. ACD-Snow model simulated a less intensive and lagged soil moisture variation in response to precipitation compared to the ACD-No-Snow

model during both the freezing and thawing period, respectively. ACD-air model affected the intensity of increased surface soil moisture, especially during the thawing period.

Three mechanisms, surface ice sublimation, snow sublimation and increased soil moisture, can contribute to enhanced latent heat flux after winter precipitation events. The relative role of each mechanism in the total mass transfer can be affected by the time and magnitude of precipitation and pre-precipitation soil

moisture/temperature states (see Sect. 4.3). The simple BCD model cannot provide a realistic partitioning of mass transfer. ACD model, with consideration of vapor diffusion and thermal effect on water flow and snowpack can produce a reasonable analysis of the relative contributions of different water flux components. With consideration of airflow, the relative contribution of each component to the mass transfer was substantially altered during the thawing period. Further work will take into account the thermal interactive

effects between snowpack and the underlying soil, which explicitly considers the convective and conductive heat fluxes and the solar radiation attenuation due to the snowpack. Such work will inevitably enhance our confidence in interpreting the underlying mechanisms and physically elaborating on the role of snowpack in cold regions.

*Code and data availability.* The coupled Soil-Snow model (STEMMUS-UEB v1.0.0) with three levels of complexity of soil water and heat transfer physics was developed based on STEMMUS-FT (Simultaneous Transfer of Energy, Momentum and Mass in Unsaturated Soils with Freeze and Thaw) and UEB (Utah Energy Balance) model. The original STEMMUS source code is available from the GitHub website via https://github.com/yijianzeng/STEMMUS. The snowmelt module is based on the code of (Tarboton and Luce, 1996). The coupled STEMMUS-UEB v1.0.0 code is archived on Zenodo (Yu et al., 2020b), licensed under the Apache License, Version 2.0. The current code is tested by MATLAB 2019b using an Intel Core i7 processor (Intel® Core™ i7-6700HQ CPU @ 2.60GHz 2.59 GHz), an installed memory (RAM, 16.0 GB), and a 64-bit Windows 10 Enterprise operating system. The relevant data can be accessed from 4TU. Center for Research Data (https://doi.org/10.4121/uuid:cc69b7f2-2448-4379-b638-09327012ce9b; https://doi.org/10.4121/uuid:61db65b1-b2aa-4ada-b41e-61ef70e57e4a, for Maqu case) and the Cold and Arid Regions Science Data Center at Lanzhou (https://doi.org/10.3972/hiwater.001.2019.db, Li, 2019, for Yakou case).

*Author contribution.* ZS, YZ, and LY designed and conceptualized this study; YZ and ZS provided the original version of STEMMUS model code and supervised the further modelling development; LY developed the STEMMUS-UEB model coupling framework with the contribution from YZ; LY and YZ prepared the original draft of the paper, LY, YZ, and ZS all contributed to the reviewing and editing of the final paper.

*Competing interests.* The authors declare that they have no conflict of interest.

### Acknowledgment

This work is supported by the National Natural Science Foundation of China (grant no. 41971033) and supported by the Fundamental Research Funds for the Central Universities, CHD (grant no. 300102298307). The authors thank the editor and referees very much for their constructive comments and suggestions on improving the manuscript.

## Appendix A

### A.1 STEMMUS-FT model with three levels of complexity

#### A.1.1 Uncoupled soil water and heat transfer physics

The Richard equation which describes the water flow under gravity and capillary forces in isothermal conditions, is solved for variably saturated soils.

$$\frac{\partial \theta}{\partial t} = -\frac{\partial q}{\partial z} - S = \rho_L \frac{\partial}{\partial z}\left[K\left(\frac{\partial \psi}{\partial z} + 1\right)\right] - S \tag{A.1}$$

where $\theta$ (m$^3$ m$^{-3}$) is the volumetric water content; $q$ (kg m$^{-2}$ s$^{-1}$) is the water flux; $z$ (m) is the vertical direction coordinate (positive upwards); $S$ (s$^{-1}$) is the sink term for root water uptake; $\rho_L$ (kg m$^{-3}$) is the soil liquid water density; $K$ (m s$^{-1}$) is the soil hydraulic conductivity; $\psi$ (m) is the soil water potential; $t$ (s) is the time.

The heat conservation equation, considering the latent heat due to water phase change, can be expressed as:

$$C_{soil}\frac{\partial T}{\partial t} - \rho_i L_f \frac{\partial \theta_i}{\partial t} = \frac{\partial}{\partial z}\left(\lambda_{eff}\frac{\partial T}{\partial z}\right) \tag{A.2}$$

where $C_{soil}$ (J kg$^{-1}$ °C$^{-1}$) is the specific heat capacity of bulk soil; $T$ (°C) is the soil temperature; $\rho_i$ (kg m$^{-3}$) is the density of soil ice; $L_f$ (J kg$^{-1}$) is the latent heat of fusion; $\theta_i$ (m$^3$ m$^{-3}$) is the soil ice volumetric water content. $\lambda_{eff}$ (W m$^{-1}$ °C$^{-1}$) is the effective thermal conductivity of the soil.

#### A.1.2 Coupled water and heat transfer

For the coupled water and heat transfer physics, the liquid water flow is non-isothermal and affected by soil temperature regimes. The movement of water vapor, as the linkage between soil water and heat flow, is explicitly characterized. With modifications made by Milly (1982), the extended version of Richards (1931) equation with consideration of the liquid and vapor flow is written as:

$$\frac{\partial}{\partial t}(\rho_L \theta_L + \rho_V \theta_V + \rho_i \theta_i) = -\frac{\partial}{\partial z}(q_L + q_V) - S$$
$$= -\frac{\partial}{\partial z}(q_{Lh} + q_{LT} + q_{Vh} + q_{VT}) - S \tag{A.3}$$
$$= \rho_L \frac{\partial}{\partial z}\left[K_{Lh}\left(\frac{\partial \psi}{\partial z} + 1\right) + K_{LT}\frac{\partial T}{\partial z}\right] + \frac{\partial}{\partial z}\left[D_{Vh}\frac{\partial \psi}{\partial z} + D_{VT}\frac{\partial T}{\partial z}\right] - S$$

where $\rho_V$ and $\rho_i$ (kg m$^{-3}$) are the density of water vapor and ice, respectively; $\theta_L$ and $\theta_V$ (m$^3$ m$^{-3}$) are the
volumetric water content (liquid and vapor, respectively); $q_L$ and $q_V$ (kg m$^{-2}$ s$^{-1}$) are the soil water fluxes of liquid water and water vapor (positive upwards), respectively. $K_{Lh}$ (m s$^{-1}$) and $K_{LT}$ (m$^2$ s$^{-1}$ °C$^{-1}$) are the isothermal and thermal hydraulic conductivities, respectively. $D_{Vh}$ (kg m$^{-2}$ s$^{-1}$) is the isothermal vapor conductivity; and $D_{VT}$ (kg m$^{-1}$ s$^{-1}$ °C$^{-1}$) is the thermal vapor diffusion coefficient.

On the basis of De Vries (1958) and Hansson et al. (2004)'s work, the heat transport function in frozen soils,
considering the fully coupled water and heat transport physics, can be expressed as:

$$\frac{\partial}{\partial t}\left[(\rho_s \theta_s C_s + \rho_L \theta_L C_L + \rho_V \theta_V C_V + \rho_i \theta_i C_i)(T - T_r) + \rho_V \theta_V L_0 - \rho_i \theta_i L_f\right] - \rho_L W \frac{\partial \theta_L}{\partial t}$$
$$= \frac{\partial}{\partial z}\left(\lambda_{eff}\frac{\partial T}{\partial z}\right) - \frac{\partial}{\partial z}[q_L C_L(T - T_r) + q_V(L_0 + C_V(T - T_r))] - C_L S(T - T_r) \tag{A.4}$$

where $C_s$, $C_L$, $C_V$ and $C_i$ (J kg$^{-1}$ °C$^{-1}$) are the specific heat capacities of solids, liquid and water vapor and ice, respectively; $\rho_s$ (kg m$^{-3}$) is the density of solids; $\theta_s$ is the volumetric fraction of solids in the soil; $T_r$ (°C) is

the arbitrary reference temperature; $L_0$ (J kg$^{-1}$) is the latent heat of vaporization of water at the reference temperature $T_r$; $W$ (J kg$^{-1}$) is the differential heat of wetting (the amount of heat released when a small amount of free water is added to the soil matrix).

**A.1.3 Coupled mass and heat physics with airflow**

In STEMMUS-FT, the temporal dynamics of three phases of water (liquid, vapor and ice), together with the soil dry air component are explicitly presented and simultaneously solved by spatially discretizing the corresponding governing equations of liquid water flow, vapor flow and airflow.

$$\frac{\partial}{\partial t}(\rho_L\theta_L + \rho_V\theta_V + \rho_i\theta_{ice}) = -\frac{\partial}{\partial z}(q_{Lh} + q_{LT} + q_{La} + q_{Vh} + q_{VT} + q_{Va}) - S$$

$$= \rho_L\frac{\partial}{\partial z}\left[K\left(\frac{\partial\psi}{\partial z} + 1\right) + D_{TD}\frac{\partial T}{\partial z} + \frac{K}{\gamma_w}\frac{\partial P_g}{\partial z}\right] + \frac{\partial}{\partial z}\left[D_{Vh}\frac{\partial\psi}{\partial z} + D_{VT}\frac{\partial T}{\partial z} + D_{Va}\frac{\partial P_g}{\partial z}\right] - S$$ (A.5)

where $q_{Lh}$, $q_{LT}$, and $q_{La}$ (kg m$^{-2}$ s$^{-1}$) are the liquid water fluxes driven by the gradient of matric potential $\frac{\partial\psi}{\partial z}$, temperature $\frac{\partial T}{\partial z}$, and air pressure $\frac{\partial P_g}{\partial z}$, respectively. $q_{Vh}$, $q_{VT}$, and $q_{Va}$ (kg m$^{-2}$ s$^{-1}$) are the water vapor fluxes driven by the gradient of matric potential $\frac{\partial\psi}{\partial z}$, temperature $\frac{\partial T}{\partial z}$, and air pressure $\frac{\partial P_g}{\partial z}$, respectively. $P_g$ (Pa) is the mixed pore-air pressure. $\gamma_W$ (kg m$^{-2}$ s$^{-2}$) is the specific weight of water; $D_{TD}$ (kg m$^{-1}$ s$^{-1}$ °C$^{-1}$) is the transport coefficient for adsorbed liquid flow due to temperature gradient; $D_{Vh}$ (kg m$^{-2}$ s$^{-1}$) is the isothermal vapor conductivity; and $D_{VT}$ (kg m$^{-1}$ s$^{-1}$ °C$^{-1}$) is the thermal vapor diffusion coefficient; $D_{Va}$ is the advective vapor transfer coefficient (Zeng et al., 2011a, b).

STEMMUS-FT takes into account different heat transfer mechanisms, including heat conduction ($\lambda_{eff}\frac{\partial T}{\partial z}$), convective heat transferred by liquid flux ($-C_Lq_L(T - T_r)$, $-C_LS(T - T_r)$), vapor flux ($-[L_0q_V + C_Vq_V(T - T_r)]$) and airflow ($q_aC_a(T - T_r)$). The latent heat of vaporization ($\rho_V\theta_VL_0$), the latent heat of freezing/thawing ($-\rho_i\theta_iL_f$) and a source term associated with the exothermic process of wetting of a porous medium (integral heat of wetting) ($-\rho_LW\frac{\partial\theta_L}{\partial t}$).

$$\frac{\partial}{\partial t}\left[(\rho_s\theta_sC_s + \rho_L\theta_LC_L + \rho_V\theta_VC_V + \rho_{da}\theta_aC_a + \rho_i\theta_iC_i)(T - T_r) + \rho_V\theta_VL_0 - \rho_i\theta_iL_f\right] -$$
$$\rho_LW\frac{\partial\theta_L}{\partial t}$$

$$= \frac{\partial}{\partial z}\left(\lambda_{eff}\frac{\partial T}{\partial z}\right) - \frac{\partial}{\partial z}[q_LC_L(T - T_r) + q_V(L_0 + C_V(T - T_r)) + q_aC_a(T - T_r)] - C_LS(T - T_r)$$ (A.6)

where $\rho_{da}$ (kg m$^{-3}$) is the density of dry air; $C_a$ (J kg$^{-1}$ °C$^{-1}$) is the specific heat capacity of dry air; $q_a$ (kg m$^{-2}$ s$^{-1}$) is the air flux. The airflow balance equation for solving the coupled water and heat equations is written as Zeng et al. (2011a, b) and Zeng and Su (2013):

$$\frac{\partial}{\partial t}[\varepsilon\rho_{da}(S_a + H_cS_L)] = \frac{\partial}{\partial z}\left[D_e\frac{\partial\rho_{da}}{\partial z} + \rho_{da}\frac{S_aK_g}{\mu_a}\frac{\partial P_g}{\partial z} - H_c\rho_{da}\frac{q_L}{\rho_L} + (\theta_aD_{Vg})\frac{\partial\rho_{da}}{\partial z}\right]$$ (A.7)

where $\varepsilon$ is the porosity; $S_a$ ($=1-S_L$) is the degree of air saturation in the soil; $S_L$ ($=\theta_L/\varepsilon$) is the degree of saturation in the soil; $H_c$ is Henry's constant; $D_e$ (m$^2$ s$^{-1}$) is the molecular diffusivity of water vapor in soil; $K_g$ (m$^2$) is the intrinsic air permeability; $\mu_a$ ( kg m$^{-2}$ s$^{-1}$) is the air viscosity; $\theta_a$ ($=\theta_V$) is the volumetric fraction of dry air in the soil; and $D_{Vg}$ (m$^2$ s$^{-1}$) is the gas phase longitudinal dispersion coefficient.

**A.2 Snowpack module UEB**

**A.2.1 Mass balance equation**

The increase or decrease of snow water equivalence with time equals the difference of income and outgoing water flux:

$$\frac{dSWE}{dt} = P_r + P_s - M_r - E \tag{A.8}$$

where $SWE$ (m) is the snow water equivalent; $P_r$ (m/s) is the rainfall rate; $P_s$ (m/s) is the snowfall rate; $M_r$ (m/s) is the meltwater outflow from the snowpack; and $E$ is the sublimation from the snowpack.

### A.2.2 Energy balance equation

The energy balance of snowpack can be expressed as:

$$\frac{dU}{dt} = Q_{sn} + Q_{li} + Q_p + Q_g - Q_{le} + Q_h + Q_e - Q_m \tag{A.9}$$

where $Q_{sn}$ (W/m$^2$) is the net shortwave radiation; $Q_{li}$ (W/m$^2$) is the incoming longwave radiation; $Q_p$ (W/m$^2$) is the advected heat from precipitation; $Q_g$ (W/m$^2$) is the ground heat flux; $Q_{le}$ (W/m$^2$) is the outgoing longwave radiation; $Q_h$ (W/m$^2$) is the sensible heat flux; $Q_e$ (W/m$^2$) is the latent heat flux due to sublimation/condensation; and $Q_m$ (W/m$^2$) is the advected heat removed by meltwater.

Equations (8) and (9) form a coupled set of first order, nonlinear ordinary differential equations. Euler predictor-corrector approach was employed in UEB model to solve the initial value problems of these equations (Tarboton and Luce, 1996).

### A.3 Albedo calculation

#### A.3.1 Ground albedo

Instead of the constant bare soil albedo in the original UEB model, the bare soil albedo is expressed as a decreasing linear function of soil moisture in STEMMUS-UEB.

$$\alpha_{g,v} = \alpha_{sat} + \min\{\alpha_{sat}, \max[(0.11 - 0.4\theta), 0]\} \tag{A.10}$$

$$\alpha_{g,ir} = 2\alpha_{g,v} \tag{A.11}$$

where $\alpha_{g,v}$ and $\alpha_{g,ir}$ are the bare soil/ground albedo for the visible and infrared band, respectively. $\alpha_{sat}$ is the saturated soil albedo, depending on local soil color. $\theta$ is the surface volumetric soil moisture.

#### A.3.2 Vegetation albedo

The calculation of vegetation albedo is developed to capture the essential features of a two-stream approximation model using asymptotic equation. It approaches the underlying surface albedo $\alpha_{g,\lambda}$ or the thick canopy albedo $\alpha_{c,\lambda}$ when the $L_{SAI}$ is close to zero or infinity.

$$\alpha_{Veg,b,\lambda} = \alpha_{c,\lambda}\left[1 - \exp\left(-\frac{\omega_\lambda \beta L_{SAI}}{\mu \alpha_{c,\lambda}}\right)\right] + \alpha_{g,\lambda} \exp\left[-\left(1 + \frac{0.5}{\mu}\right)L_{SAI}\right] \tag{A.12}$$

$$\alpha_{Veg,d,\lambda} = \alpha_{c,\lambda}\left[1 - \exp\left(-\frac{2\omega_\lambda \beta L_{SAI}}{\alpha_{c,\lambda}}\right)\right] + \alpha_{g,\lambda} \exp[-2\,L_{SAI}] \tag{A.13}$$

where subscripts $Veg, b, d, c, g$ and $\lambda$ represent vegetation, direct beam, diffuse radiation, thick canopy, ground, and spectrum bands of either visible or infrared bands. $\mu$ is the cosine of solar zenith angle; $\omega_\lambda$ is the single scattering albedo, 0.15 for visible and 0.85 for infrared band, respectively; $\beta$ is assigned as 0.5; $L_{SAI}$ is the sum of leaf area index LAI and stem area index SAI; $\alpha_{c,\lambda}$ is the thick canopy albedo dependent on vegetation types.

The bulk snow-free surface albedo, averaged between bare ground albedo and vegetation albedo, then is written as:

$$\alpha_{\eta,\lambda} = \alpha_{Veg,\lambda} f_{Veg} + \alpha_{g,\lambda}(1 - f_{Veg}) \tag{A.14}$$

where $\alpha_{\eta,\lambda}$ is the averaged bulk snow-free surface albedo; $f_{Veg}$ is the fraction of vegetation cover.

### A.3.3 Snow albedo

According to Dickinson et al. (1993), snow albedo can be expressed as a function of snow surface age and solar illumination angle. The snow surface age, which is dependent on snow surface temperature and snowfall, is updated with each time step in UEB. Visible and near infrared bands are separately treated when calculating reflectance, which are further averaged as the albedo with modifications of illumination angle and snow age. The reflectance in the visible and near infrared bands can be written as:

$$\alpha_{vd} = (1 - C_v S_{age})\alpha_{vo} \tag{A.15}$$

$$\alpha_{ird} = (1 - C_{ir} S_{age})\alpha_{iro} \tag{A.16}$$

where $\alpha_{vd}$ and $\alpha_{ird}$ represent diffuse reflectance in the visible and near infrared bands, respectively. $C_v$ (= 0.2) and $C_{ir}$ (=0.5) are parameters that quantify the sensitivity of the visible and infrared band albedo to snow surface aging (grain size growth), $\alpha_{vo}$ (=0.85) and $\alpha_{iro}$ (=0.65) are fresh snow reflectance in visible and infrared bands, respectively. $S_{age}$ is a function to account for aging of the snow surface, and is given by:

$$S_{age} = \frac{\tau}{1 + \tau} \tag{A.17}$$

where $\tau$ is the non-dimensional snow surface age that is incremented at each time step by the quantity designed to emulate the effect of the growth of surface grain sizes.

$$\Delta\tau = \frac{r_1 + r_2 + r_3}{\tau_o} \Delta t \tag{A.18}$$

where $\Delta t$ is the time step in seconds with $\tau_o = 10^6$ s. $r_1$ is the parameter to represent the effect of grain growth due to vapor diffusion, and is dependent on snow surface temperature:

$$r_1 = \exp\left[5000\left(\frac{1}{273.16} - \frac{1}{T_s}\right)\right] \tag{A.19}$$

$r_2$ describes the additional effect near and at the freezing point due to melt and refreeze:

$$r_2 = \min(r_1^{10}, 1) \tag{A.20}$$

$r_3$=0.03 (0.01 in Antarctica) represents the effect of dirt and soot.

The reflectance of radiation with illumination angle (measured relative to the surface normal) is computed as:

$$\alpha_v = \alpha_{vd} + 0.4 f(\varphi)(1 - \alpha_{vd}) \tag{A.21}$$

$$\alpha_{ir} = \alpha_{ird} + 0.4\, f(\varphi)(1 - \alpha_{ird}) \tag{A.22}$$

where $f(\varphi) = \begin{cases} \frac{1}{b}\left[\frac{b+1}{1+2b\cos(\varphi)} - 1\right], & for \cos(\varphi) < 0.5 \\ 0, & otherwise \end{cases}$

where $b$ is a parameter set at 2 as Dickinson et al. (1993).

When the snowpack is shallow (depth z<h=0.01m), the albedo is calculated by interpolating between the snow albedo and bare ground albedo with the exponential term approximating the exponential extinction of radiation penetration of snow.

$$A_{v/ir} = r\alpha_{g,v/ir} + (1 - r)\alpha_{v/ir} \tag{A.23}$$

where $r = \left(1 - \frac{z}{h}\right)e^{-z/2h}$.

**Appendix B**

**B.1 Snow water equivalent**

STEMMUS-UEB can reproduce the dynamics of snow water equivalent (Figure B1). The discrepancies were mainly happened under conditions of the less snow water equivalent. These intermitted shallow snowpack processes are difficult to be well captured, due to the drifting snow effect, temporal and complex ground heat conditions, requires both the high-quality observations and advanced snowpack models.

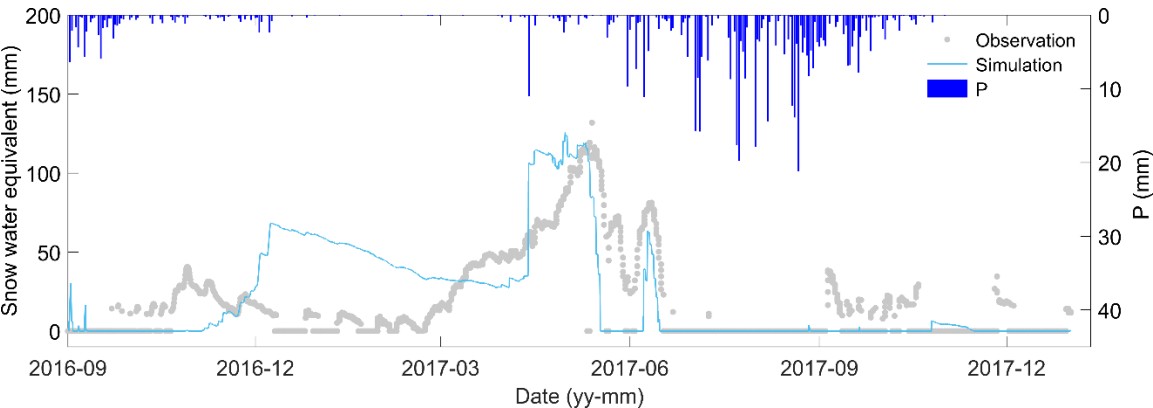

**Figure B1. Time series of the observed and estimated snow water equivalent using the developed STEMMUS-UEB model.**

**B.2 Daily surface evaporation**

Compared to the observation, surface evaporation was underestimated by the model with no snow module during the snowfall periods (Figure B2). Model with snow module, however, produced a general good agreement but with overestimations and underestimations, which corresponds to the mismatches in the snow

water equivalent results. When the snow water equivalent is overestimated, snowpack sublimation and thus the surface evaporation was overestimated.

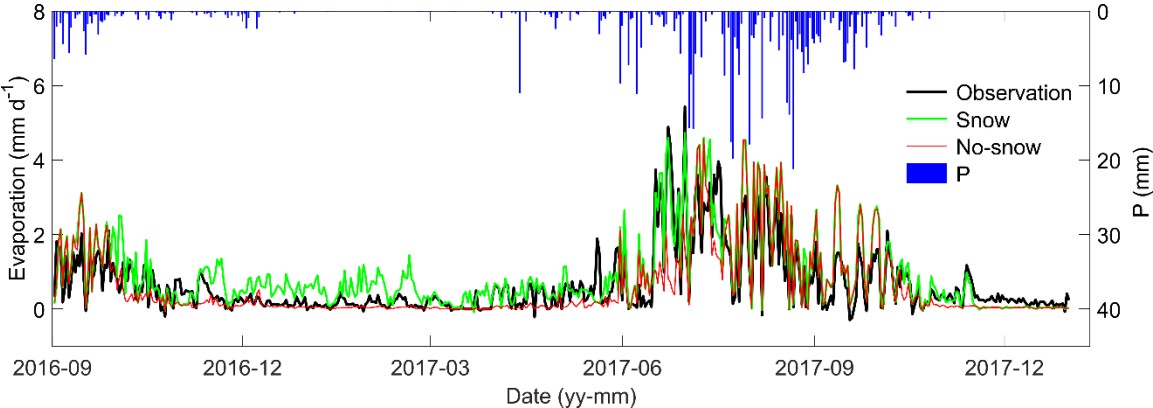

**Figure B2. Intercomparison of the observed and estimated surface evaporation using the model with and without the snow module.**

Compared to the model without snow module, the model with snow module produced a better correlation to the measured daily surface evaporation (Figure B3). Surface evaporation was underestimated by the model without snow module while slightly overestimated by the model with snow module.

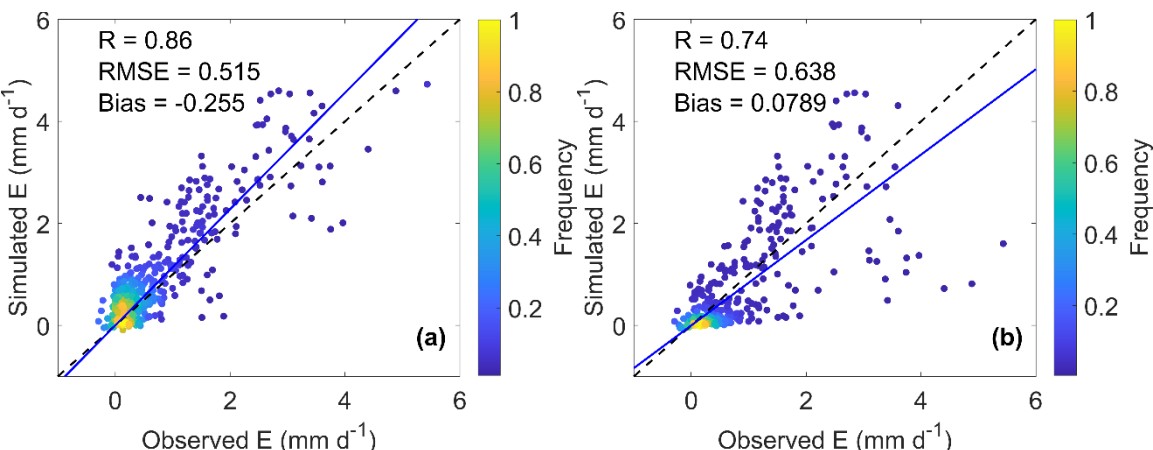

**Figure B3. Measured and estimated daily surface evaporation using the model with and without snow**
**module (a and b, respectively).**

**B.3 Soil moisture and temperature**

Both models with and without snow module can reproduce the soil moisture dynamics well with the response to precipitation events (Figure B4). Soil moisture was underestimated by the model without snow module due to less amount of incoming water flux. Such underestimation was damped as the soil depth increases.
Model with snow module gains more incoming water (snowmelt water) and the underestimation of soil moisture was alleviated.

The dynamics of soil temperature was well reproduced by models with and without snow module (Figure B5). There is no significant difference in soil temperature simulations between models with and without snow module.

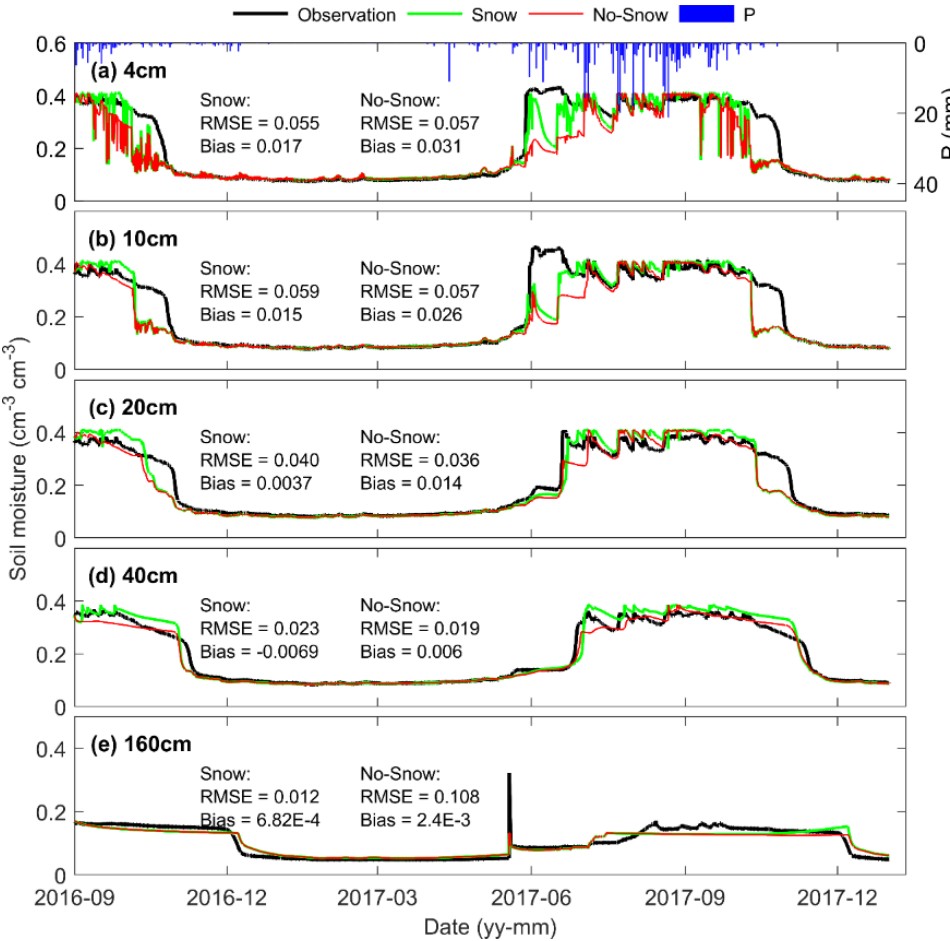

**Figure B4. Observed and estimated soil moisture at various soil layers using the model with and without the snow module.**

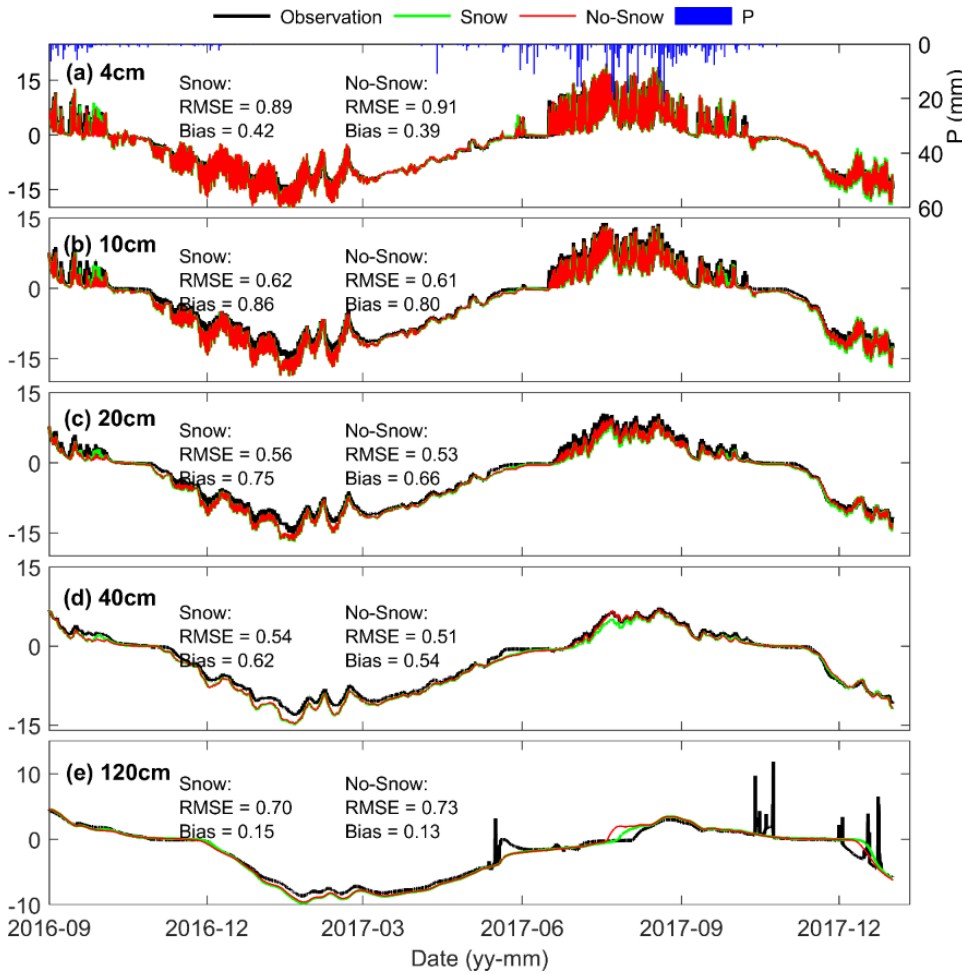

**Figure B5. Observed and estimated soil temperature at various soil layers using the model with and without the snow module.**

**B.4 Snow cover properties and albedo**

There is a good correlation between the snow depth and surface albedo (Figure B6). Figure B7 shows that surface albedo variations correspond well to the dynamics of the snow cover properties. This demonstrated that surface albedo is a reliable indicator to identify the presence of the snowpack and its influencing periods. Three example periods were selected to illustrate the validity of using the indirect method (the albedo variation, ancillary meteorological data: air temperature, and precipitation) to define the presence and lasting time of the snowpack. Results indicated that the snowpack duration was successfully characterized using the indirect method (results were shown in the Supplement II Table S4).

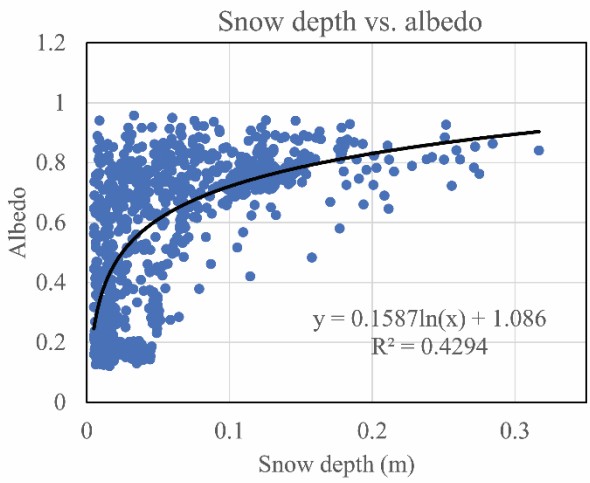

**Figure B6. Scatter plot of the snow depth and albedo (Yakou station, 2014-2017).**

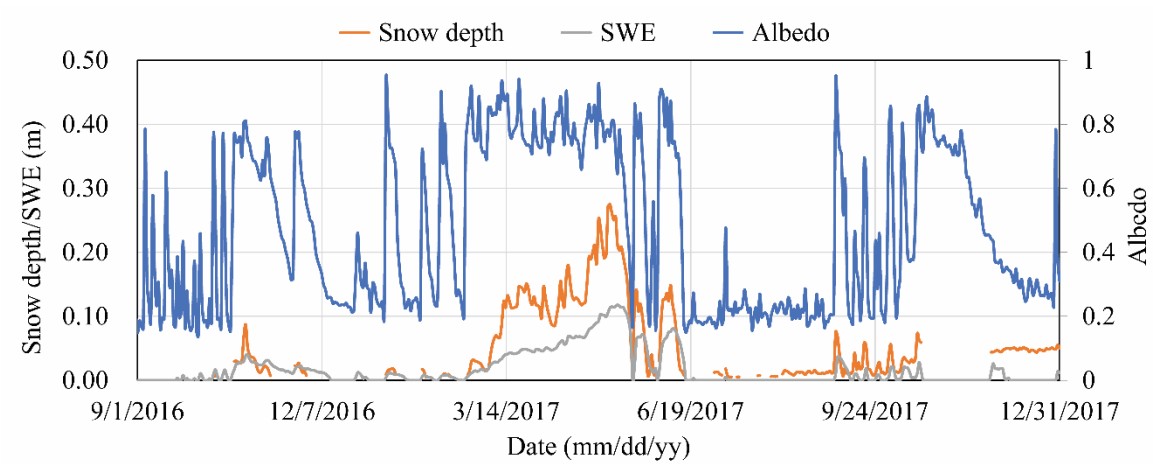

**Figure B7. Time series of the snow depth, snow water equivalent (SWE), and albedo.**

## Notation

| Symbol | Parameter | Unit | Value |
|--------|-----------|------|-------|
| **Main inputs** | | | |
| Soil model component (STEMMUS-FT) | | | |
| $a$ | Fitted parameter for soil surface resistance | - | 0.3565 |
| $b(z)$ | Normalized water uptake distribution | $m^{-1}$ | |
| $C_a$ | Specific heat capacity of dry air | $J\,kg^{-1}\,{}^{\circ}C^{-1}$ | 1.005 |
| $C_{app}$ | Apparent heat capacity | $J\,kg^{-1}\,{}^{\circ}C^{-1}$ | |
| $C_i$ | Specific heat capacity of ice | $J\,kg^{-1}\,{}^{\circ}C^{-1}$ | 2.0455 |
| $C_L$ | Specific heat capacity of liquid | $J\,kg^{-1}\,{}^{\circ}C^{-1}$ | 4.186 |
| $C_s$ | Specific heat capacity of soil solids | $J\,kg^{-1}\,{}^{\circ}C^{-1}$ | |
| $C_{soil}$ | Heat capacity of the bulk soil | $J\,kg^{-1}\,{}^{\circ}C^{-1}$ | |
| $C_V$ | Specific heat capacity of water vapor | $J\,kg^{-1}\,{}^{\circ}C^{-1}$ | 1.87 |
| $c_p$ | Specific heat capacity of air | $J\,kg^{-1}\,K^{-1}$ | |
| $D_e$ | Molecular diffusivity of water vapor in soil | $m^2\,s^{-1}$ | |
| $D_{TD}$ | Transport coefficient for adsorbed liquid flow due to temperature gradient | $kg\,m^{-1}\,s^{-1}\,{}^{\circ}C^{-1}$ | |
| $D_{Va}$ | Advective vapor transfer coefficient | s | |
| $D_{Vg}$ | Gas phase longitudinal dispersion coefficient | $m^2\,s^{-1}$ | |
| $D_{Vh}$ | Isothermal vapor conductivity | $kg\,m^{-2}\,s^{-1}$ | |
| $D_{VT}$ | Thermal vapor diffusion coefficient | $kg\,m^{-1}\,s^{-1}\,{}^{\circ}C^{-1}$ | |
| $H_c$ | Henry's constant | - | 0.02 |
| $K$ | Hydraulic conductivity | $m\,s^{-1}$ | |
| $K_g$ | Intrinsic air permeability | $m^2$ | |
| $K_{Lh}$ | Isothermal hydraulic conductivities | $m\,s^{-1}$ | |
| $K_{LT}$ | Thermal hydraulic conductivities | $m^2\,s^{-1}\,{}^{\circ}C^{-1}$ | |
| $K_s$ | Soil saturated hydraulic conductivity | $m\,s^{-1}$ | |
| $L_0$ | Latent heat of vaporization of water at the reference temperature | $J\,kg^{-1}$ | |
| $LAI_{eff}$ | Effective leaf area index | - | |
| $L_f$ | Latent heat of fusion | $J\,kg^{-1}$ | 3.34E+05 |
| $n$ | Van Genuchten fitting parameters | - | |
| $r_a^c$ | Aerodynamic resistance for canopy surface | $s\,m^{-1}$ | |
| $r_a^s$ | Aerodynamic resistance for bare soil | $s\,m^{-1}$ | |
| $r_{c,min}$ | Minimum canopy surface resistance | $s\,m^{-1}$ | |
| $r_{l,min}$ | Minimum leaf stomatal resistance | $s\,m^{-1}$ | |
| $r_s$ | Soil surface resistance | $s\,m^{-1}$ | |
| $r_{sl}$ | Resistance to molecular diffusion of the water surface | $s\,m^{-1}$ | 10 |
| $R_n$ | Net radiation | $MJ\,m^{-2}\,day^{-1}$ | |
| $R_n^c$ | Net radiation at the canopy surface | $MJ\,m^{-2}\,day^{-1}$ | |
| $R_n^s$ | Net radiation at the soil surface | $MJ\,m^{-2}\,day^{-1}$ | |
| $S_a$ | Degree of saturation of the soil air | - | $=1-S_L$ |
| $S_L$ | Degree of water saturation in the soil | - | $=\theta_L/\varepsilon$ |
| $S_p$ | Potential water uptake rate | $s^{-1}$ | |

| | | | |
|---|---|---|---|
| $t$ | Time | s | |
| $T_p$ | Potential transpiration | m s$^{-1}$ | |
| $T_r$ | Arbitrary reference temperature | °C | 20 |
| $W$ | Differential heat of wetting | J kg$^{-1}$ | |
| $z$ | Vertical space coordinate (positive upwards) | m | |
| $\alpha$ | Air entry value of soil | m$^{-1}$ | |
| $a(h)$ | Reduction coefficient related to soil water potential | - | |
| $\varepsilon$ | Porosity | - | |
| $\lambda_{eff}$ | Effective thermal conductivity of the soil | W m$^{-1}$ $^{°C-1}$ | |
| $\theta_s$ | Volumetric fraction of solids in the soil | m$^3$ m$^{-3}$ | |
| $\theta_{sat}$ | Saturated soil water content | m$^3$ m$^{-3}$ | |
| $\theta_r$ | Residual soil water content | m$^3$ m$^{-3}$ | |
| $\theta_l$ | Topsoil water content | m$^3$ m$^{-3}$ | |
| $\theta_{min}$ | Minimum water content above which soil is able to deliver vapor at a potential rate | m$^3$ m$^{-3}$ | |
| $\rho_a$ | Air density | kg m$^{-3}$ | |
| $\rho_{da}$ | Density of dry air | kg m$^{-3}$ | |
| $\rho_i$ | Density of ice | kg m$^{-3}$ | 920 |
| $\rho_L$ | Density of soil liquid water | kg m$^{-3}$ | 1000 |
| $\rho_s$ | Density of solids | kg m$^{-3}$ | |
| $\rho_V$ | Density of water vapor | kg m$^{-3}$ | |
| $\gamma_W$ | Specific weight of water | kg m$^{-2}$ s$^{-2}$ | |
| $\mu_a$ | Air viscosity | kg m$^{-2}$ s$^{-1}$ | |

**Snow model component (UEB)**

| | | | |
|---|---|---|---|
| $T_r$ | Air temperature above which precipitation is all rain | °C | |
| $T_{sn}$ | Air temperature below which precipitation is all snow | °C | |
| $\varepsilon_{sn}$ | Emissivity of snow | - | |
| $C_g$ | Ground heat capacity | J kg$^{-1}$ $^{°C-1}$ | |
| $z_o$ | Snow surface aerodynamic roughness | m | |
| $L_c$ | Liquid holding capacity of snow | - | |
| $K_{sn}$ | Snow saturated hydraulic conductivity | m h$^{-1}$ | |
| $\alpha_{vo}$ | Visual new snow albedo | - | |
| $\alpha_{iro}$ | Near-infrared new snow albedo | - | |
| $\alpha_{bg}$ | Bare ground albedo | - | Eqs. A10 - A14 |
| $D_e$ | Thermally active depth of soil | m | |
| $\lambda_{sn}$ | Snow surface thermal conductivity | m h$^{-1}$ | |
| $\rho_{sn}$ | Snow density | kg m$^{-3}$ | |
| $A_{ed}$ | Albedo extinction depth | m | |
| $F_c$ | Forest cover fraction | - | |
| $D_f$ | Drift factor | - | |
| $\rho_s$ | Soil density | kg m$^{-3}$ | |

**Main outputs**

**Soil model component (STEMMUS-FT)**

| | | | |
|---|---|---|---|
| $\psi$ | Soil water potential | m | |

| | | |
|---|---|---|
| $P_g$ | Mixed pore-air pressure | Pa |
| $T$ | Soil temperature | °C |
| $\theta$ | Volumetric water content | $\text{m}^3\,\text{m}^{-3}$ |
| $\theta_i$ | Soil ice volumetric water content | $\text{m}^3\,\text{m}^{-3}$ |
| $\theta_L$ | Soil liquid volumetric water content | $\text{m}^3\,\text{m}^{-3}$ |
| $\theta_V$ | Soil vapor volumetric water content | $\text{m}^3\,\text{m}^{-3}$ |
| $\theta_a$ | Volumetric fraction of dry air in the soil | $\text{m}^3\,\text{m}^{-3}$ |
| $q$ | Water flux | $\text{kg m}^{-2}\,\text{s}^{-1}$ |
| $q_a$ | Dry air flux | $\text{kg m}^{-2}\,\text{s}^{-1}$ |
| $q_L$ | Soil liquid water fluxes (positive upwards) | $\text{kg m}^{-2}\,\text{s}^{-1}$ |
| $q_{La}$ | Liquid water flux driven by the gradient of air pressure | $\text{kg m}^{-2}\,\text{s}^{-1}$ |
| $q_{Lh}$ | Liquid water flux driven by the gradient of matric potential | $\text{kg m}^{-2}\,\text{s}^{-1}$ |
| $q_{LT}$ | Liquid water flux driven by the gradient of temperature | $\text{kg m}^{-2}\,\text{s}^{-1}$ |
| $q_V$ | Soil water vapor fluxes (positive upwards) | $\text{kg m}^{-2}\,\text{s}^{-1}$ |
| $q_{Va}$ | Water vapor flux driven by the gradient of air pressure | $\text{kg m}^{-2}\,\text{s}^{-1}$ |
| $q_{Vh}$ | Water vapor flux driven by the gradient of matric potential | $\text{kg m}^{-2}\,\text{s}^{-1}$ |
| $q_{VT}$ | Water vapor flux driven by the gradient of temperature | $\text{kg m}^{-2}\,\text{s}^{-1}$ |
| $S$ | Sink term for transpiration | $\text{s}^{-1}$ |
| $S_h$ | Latent heat flux density | $\text{W m}^{-3}$ |
| Snow model component (UEB) | | |
| $P_r$ | Precipitation in the form of rain | $\text{m s}^{-1}$ |
| $P_s$ | Precipitation in the form of snow | $\text{m s}^{-1}$ |
| $SWE$ | Snow water equivalent | m |
| $Q_h$ | Surface Sensible Heat Flux | $\text{W m}^{-2}$ |
| $Q_e$ | Surface Latent Heat Flux | $\text{W m}^{-2}$ |
| $E$ | Surface Sublimation | $\text{m s}^{-1}$ |
| $T_{surf}$ | Snow Surface Temperature | °C |
| $U$ | Energy Content | |
| $M_r$ | Melt outflow rate | $\text{m s}^{-1}$ |
| $A_{v/ir}$ | Surface Albedo | - |
| $Q_m$ | Heat advected by melt outflow | $\text{W m}^{-2}$ |
| $Q_{sn}$ | Net shortwave radiation | $\text{W m}^{-2}$ |
| $Q_{li}$ | Net longwave radiation | $\text{W m}^{-2}$ |
| $\tau$ | No-dimensional snow age | - |

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

**Tables and Figures**

**Table 1. Brief overview of current soil-snow modelling efforts.**

| Model | Soil | | | | | Snow | | | | | | Relevant reference |
|---|---|---|---|---|---|---|---|---|---|---|---|---|
| | Water balance | Energy balance | Air balance | Water-heat coupled | Others (vapor, freeze-thaw, convective heat) | Snow layer | Snow energy budget | Water flow | Snow albedo | Snow density | Other processes (snow compaction, wind, and vegetation effect) | |
| CABLE-SLI | Richards | HT_cond, Advc | No | Yes | Vapor; HT_convect (liquid) | Multilayer | HT_cond, Advc | Mass conservation | Albedo_SNW_1A | Density_SNW_1 | Snow compaction (overburden and metamorphism) | Cuntz and Haverd (2018) |
| CLASS | Richards | HT_cond | No | No | No vapor; LH_phas | Single | HT_cond | Snowfall, energy driven snow melting | Albedo_SNW_1B | Density_SNW_2A | - | Barlett et al. (2006) |
| CLM5 | Richards | HT_cond | No | No | No vapor; LH_phas | Multilayer up to five | HT_cond | Mass conservation | Albedo_SNW_2 | Density_SNW_4A | Snow compaction (metamorphism, overburden, melting, wind-drift) | Lawrence et al., (2019) |
| HTESSEL | Richards | HT_cond | No | No | No vapor; LH_phas | Single | HT_cond | Mass conservation | Albedo_SNW_3B | Density_SNW_4B | Snow compaction (overburden and metamorphism) | Dutra et al. (2010) |
| HTESSEL-ML | Richards | HT_cond | No | No | No vapor; LH_phas | Multilayer up to 3 | HT_cond | Mass conservation | Albedo_SNW_3B | Density_SNW_4B | Snow compaction (overburden and metamorphism) | Dutra et al. (2012) |
| SURFEX-ISBA-ES01 | Richards | HT_cond | No | No | No vapor; LH_phas | Multilayer, 3 | HT_cond | Mass conservation | Albedo_SNW_3A | Density_SNW_4C | Snow compaction and settling | Boone and Etchevers (2001) |
| SURFEX-ISBA-ES16 | Richards | HT_cond | No | No | No vapor; LH_phas | Multilayer, 12 | HT_cond | Mass conservation | Albedo_SNW_3C | Density_SNW_4D | Snow compaction; wind-induced densification | Decharme et al. (2016) |
| SURFEX-ISBA-MEB | Richards | HT_cond | No | No | No vapor; LH_phas | Multilayer, 12 | HT_cond | Mass conservation | Albedo_SNW_3C | Density_SNW_4D | Snow compaction; wind-induced densification; Vegetation effect (interception/ unloading; snow fraction); litter layer; Multi-component energy balance | Boone et al. (2017) |
| SURFEX-Crocus | Richards | HT_cond | No | No | No vapor; LH_phas | Multilayer (dynamic) | HT_cond | Mass conservation | Albedo_SNW_3D | Density_SNW_4F | Snow metamorphism; compaction; wind drift; sublimation/ hoar deposition | Vionnet et al. (2012) |
| JSBACH | Richards | HT_cond | No | No | No vapor; LH_phas | Multilayer up to 5 | HT_cond | Mass conservation | Constant | Constant | - | Ekici et al. (2014) |
| JULES | Richards | HT_cond | No | No | No vapor; LH_phas | Multilayer up to 5 | HT_cond | Mass conservation | Albedo_SNW_3A | Density_SNW_4B | Snow compaction | Best (2011) |
| Noah-MP | Richards | HT_cond | No | No | No vapor; LH_phas | Multilayer up to 3 | HT_cond | Mass conservation | Albedo_SNW_2 | Density_SNW_2B | - | Niu et al. (2011) |
| ORCHIDEE-ES | Richards | HT_cond | No | No | No vapor; LH_phas | Multilayer, 3 | HT_cond | Mass conservation | Albedo_SNW_3E | Density_SNW_4B | Snow compaction (overburden and metamorphism) | Wang et al. (2013) |
| SNOWPACK | Richards | HT_cond | No | Yes | Vapor; HT_convect (liquid) | Multilayer | HT_cond | Mass conservation, vapor | Albedo_SNW_3D | Density_SNW_4G | Explicit prognostic settlement; Snow metamorphism; compaction; wind drift; sublimation | Lehning et al. (1999) |
| WEB-DHM | Richards | HT_cond | No | No | No vapor; LH_phas | Single | HT_cond | Mass conservation | Albedo_SNW_1B | Constant | Vegetation interception | Wang et al. (2009) |
| WEB-DHM-S | Richards | HT_cond | No | No | No vapor; LH_phas | Multilayer up to 3 | HT_cond | Mass conservation | Albedo_SNW_3F | Density_SNW_4B | Snow compaction | Shrestha et al. (2010) |
| HydroSiB2-SF | Richards | HT_cond | No | Yes | Vapor; enthalpy-based FT; LH_phas | Multilayer up to 3 | HT_cond, Advc | Mass conservation | Albedo_SNW_3F | Density_SNW_4B | Snow compaction | Wang et al. (2017) |
| WEB-GM | - | - | - | - | - | Multilayer, vary with snow depth | Enthalpy based heat transfer | Mass conservation | Albedo_SNW_4 | Density_SNW_3 | Snow compaction (metamorphism, snow densification, melting); | Ding et al. (2017) |

| Model | Water flow | Heat transfer | | | Vapor | Snow layer | Snow heat | Mass | Snow albedo | Snow density | Snow processes | Reference |
|---|---|---|---|---|---|---|---|---|---|---|---|---|
| SWAP | Richards | HT_cond | No | No | No vapor; LH_phas | Single | - | Mass conservation | Constant | Density_SNW_4H | Vegetation interception | Gusev and Nasonova (2003) |
| COUP | Richards | HT_cond, Advc | No | Yes | Vapor; HT_convect (liquid) | Single | HT_cond | Mass conservation | Albedo_SNW_1A | Density_SNW_2C | Snow compaction | Jansson (2012) |
| SHAW | Richards | HT_cond, Advc | No | Yes | Vapor; HT_convect (liquid, vapor) | Multilayer | HT_cond, Advc | Mass conservation, vapor | Albedo_SNW_1C | Density_SNW_4E | Snow compaction, settling | Flerchinger and Saxton (1989); Flerchinger (2017) |
| HYDRUS | Richards | HT_cond, Advc | No | Yes | Vapor; HT_convect (liquid, vapor) | - | - | - | - | - | - | Hansson et al. (2004); Šimůnek et al. (2008) |
| STEMMUS-UEB | Richards | HT_cond, Advc | Yes | Yes | Vapor; LH_phas; HT_convect (liquid, vapor, dry air); Various complexity of SHP | Single | HT_cond, Advc | Mass conservation | Albedo_SNW_3F | Constant | Empirical wind drift and vegetation interception | This study |

Note:

HT_cond, Heat conduction;

Advc, Advection;

LH_phas, Latent heat due to phase change;

HT_Convect, Convective heat due to liquid;

SHP, soil physical process;

Albedo_SNW_1A, Snow albedo 1A, Function of snow age;

Albedo_SNW_1B, Snow albedo 1B, Empirical function, considering dry/wet states;

Albedo_SNW_1C, Snow albedo 1C, Function of extinction coefficient, grain-size, and solar zenith angle;

Albedo_SNW_2, Snow albedo 2, Two-stream radiative transfer solution, considering snow aging, solar zenith angle, optical parameters, 975 and impurity;

Albedo_SNW_3A, Snow albedo 3A, Prognostic snow albedo, considering aging effect;

Albedo_SNW_3B, Snow albedo 3B, Prognostic snow albedo, considering aging effect and vegetation type dependent;

Albedo_SNW_3C, Snow albedo 3C, Prognostic snow albedo, considering aging and optical diameter;

Albedo_SNW_3D, Snow albedo 3D, Prognostic snow albedo, considering age and microstructure;

Albedo_SNW_3E, Snow albedo 3E, Prognostic snow albedo, considering aging effect and dry/wet states;

Albedo_SNW_3F, Snow albedo 3F, Prognostic snow albedo considering aging effect, solar zenith angle;

Albedo_SNW_4, Snow albedo 4, Diagnostic snow albedo, considering snow aging, sleet/snowfall fraction, grain diameter, cloud fraction, and solar elevation effect;

Density_SNW_1, Snow density 1, relying on in situ measurements;

Density_SNW_2A, Snow density 2A, function of air temperature;

Density_SNW_2B, Snow density 2B, Function of extinction coefficient and grain-size;

Density_SNW_2C, Snow density 2C, Function of old (densification), new-fallen (air temperature) snow pack density, and snow depth;

Density_SNW_3, Snow density 3, Diagnostic density, considering wet-bulb temperature;

Density_SNW_4A, Snow density 4A, Prognostic density, considering temperature, wind effect, snow compaction, water/ice states;

Density_SNW_4B, Snow density 4B, Prognostic density, considering overburden and thermal metamorphisms;

Density_SNW_4C, Snow density 4C, Prognostic snow density, considering snow compaction and settling;

Density_SNW_4D, Snow density 4D, Prognostic snow density, considering snow compaction and wind-induced densification;

Density_SNW_4E, Snow density 4E, Prognostic snow density, considering snow compaction, settling, and vapor transfer;

Density_SNW_4F, Snow density 4F, Prognostic density, function of wind speed and air temperature;

Density_SNW_4G, Snow density 4G, Prognostic density, function of stress state and microstructure;

Density_SNW_4H, Snow density 4H, Prognostic density, considering snow temperature.

**Table 2. Main subroutines in STEMMUS-UEB**

| Model Subroutines | Main functions | Main inputs | Main outputs | Subroutine-Connections |
|---|---|---|---|---|
| **Soil module** | | | | |
| Air_sub | Solves soil dry air balance equation | Water vapor density, diffusivity, dispersion coefficient; dry air density, gas conductivity, flux; liquid water flux; top and bottom boundary conditions | Soil air pressure profile | CondV_DVg, CondL_h, Condg_k_g, Density_V, h_sub --->; --> Enrgy_sub, |
| CondL_h | Calculates soil hydraulic conductivity | Soil hydraulic parameters; soil matric potential; soil temperature | Soil hydraulic conductivity; soil water content | StartInit --->; --> h_sub; Air_sub; Enrgy_sub, |
| CondT_coeff | Calculates soil thermal capacity and conductivity | Thermal properties of soil constituents; soil texture; soil water content; volumetric fraction of dry air; dry air density; vapor density | Soil thermal capacity and conductivity | StartInit, CondL_h, Density_V, Density_DA, EfeCapCond --->; --> Enrgy_sub, |
| CondV_DVg | Calculates flux of dry air and vapor dispersity | Gas conductivity, dry air pressure, volumetric fraction of dry air; saturated soil water content | Dry air flux and vapor dispersion coefficient | StartInit, CondL_h, Condg_k_g --->; --> h_sub; Air_sub; Enrgy_sub, |
| CondL_Tdisp | Calculates transport coefficient for adsorbed liquid flow | Soil porosity, soil water content, temperature, matric potential, volumetric fraction of dry air | Transport coefficient for adsorbed liquid flow and the heat of wetting | StartInit, CondL_h, Condg_k_g --->; --> h_sub; Enrgy_sub, |
| Condg_k_g | Calculates gas conductivity | Soil porosity, saturated hydraulic conductivity, volumetric fraction of dry air | Gas conductivity | StartInit, CondL_h --->; --> CondV_DVg, |
| Density_DA | Calculates dry air density | Soil temperature, matric potential, dry air pressure; vapor density and its derivative with respect to temperature and matric potential | Density of dry air | StartInit, CondL_h, Density_V --->; --> CondT_coeff, Air_sub, Enrgy_sub, |
| Density_V | Calculates vapor density and its derivative with respect to temperature and matric potential | Soil temperature, matric potential | Vapor density and its derivative with respect to temperature and matric potential | CondL_h --->; --> Density_DA, CondT_coeff, h_sub, Air_sub, Enrgy_sub, |
| EfeCapCond | Calculates soil thermal capacity and conductivity | Thermal properties of soil constituents; soil texture; soil water content; volumetric fraction of dry air; dry air density; vapor density | Soil heat capacity; thermal conductivity | StartInit, CondL_h, Density_V, Density_DA --->; --> CondT_coeff, |
| Enrgy_sub | Solves soil energy balance equation | Soil thermal properties, soil hydraulic conductivity, soil matric potential, soil water content, soil temperature, soil dry air pressure, density of dry air, heat of wetting, vapor density, liquid water flux, vapor flux, dry air flux, meterological forcing, top and bottom boundary conditions | Soil temperature profile, liquid water flux, vapor flux, and dry air flux, surface and bottom energy fluxes | Air_sub, h_sub, CondL_h, CondV_DVg, CondL_Tdisp, CondT_coeff, Density_D, Density_DA, PREDICORR --->, |
| Forcing_PARM | Disaggregates the meteorological forcing into the required time steps | Observed meteorological forcing at hourly/daily time scale | Meteorological forcings at model required time scale | StartInit --->; --> h_sub, Enrgy_sub, |
| h_sub | Solves soil water balance equation | Soil temperature, soil water content, matric potential, soil hydraulic conductivity, heat of wetting, soil dry air pressure, vapor density, diffusivity, dispersity, volumetric fraction of vapor, meteorological forcing, top and bottom boundary conditions | Soil matric potential profile, top and bottom water fluxes, evaporation | StartInit, CondV_DVg, CondL_h, CondV_DE, CondL_Tdisp, Condg_k_g, Density_V, Forcing_PARM, ALBEDO, PARTSNOW, PREDICORR --->; --> Air_sub, Enrgy_sub, |
| StartInit | Initializes model setup | Soil texture, thermal properties of soil constituents, initial soil water content and temperature, top and bottom boundary condition settings | - | --> CondV_DVg, CondL_h, CondV_DE, CondL_Tdisp, Condg_k_g, Density_DA, EfeCapCond, Forcing_PARM, h_sub, |
| Diff_Moisture_Heat | Solves soil water and energy balance equations independently | Soil thermal properties, soil hydraulic conductivity, soil matric potential, soil water content, soil temperature, meteorological forcing, top and bottom boundary conditions | Soil water content and temperature profile, liquid water flux, surface and bottom water and energy fluxes | StartInit, CondT_coeff, Forcing_PARM, ALBEDO, PARTSNOW, PREDICORR --->, |
| **Snowpack module** | | | | |
| agesn | Calculates snow age | Snow surface temperature, snowfall | Updated snow age | PARTSNOW, PREDICORR --->; --> ALBEDO, |
| ALBEDO | Calculates snow albedo | Fresh snow reflectance at visible and near infrared bands, snow age, bare ground albedo, albedo extinction parameter, snow water equivalent | Snow albedo | agesn --->; --> PREDICORR, |
| PARTSNOW | Partitions precipitation into rainfall and snowfall | Precipitation, air temperature, temperature thresholds for rainfall/snowfall | Rainfall, snowfall | Forcing_PARM --->; --> PREDICORR, |

| | | | | |
|---|---|---|---|---|
| PREDICORR | Solves the snow mass and energy balance equations and updates state variables SWE and U | Air temperature, snow albedo, wind speed, relative humidity, rainfall/snowfall, shortwave/longwave radiation, site parameters | Snow energy content, water equivalent, snow albedo, snow surface temperature, meltwater outflow rate, snow sublimation, snowfall/rainfall | Forcing_PARM --->; --> agesn[2], ALBEDO[2]. |

Note:

---> means the relevant subroutines which are incoming to the current one, --> means the relevant subroutines for which the current subroutine is output to;

agesn[2] and ALBEDO[2], means the use of subroutines agesn and ALBEDO after solving the snowpack energy and mass conservation equations, to update the snow age and albedo.

**Table 3. Numerical experiments with various mass and energy transfer schemes with/without explicit consideration of snow cover (Eqs. A1-A7 are listed in Appendix A.1; Eqs. A8-A9 are listed in Appendix A.2).**

| Processes | | Experiments | |
|---|---|---|---|
| Snowpack (SNW) | Mass and energy transfer in soils (SMETr) | | |
| SNW =1: UEB (Eqs. A.8 & A.9) | SMETr=1: basic coupled water-heat transfer (Eqs. A.1 & A.2) | BCD-Snow | STEMMUS-FT_Snow |
| | SMETr=2: advanced coupled water-heat transfer without airflow (Eqs. A.3 & A.4) | ACD-Snow | |
| | SMETr=3: advanced coupled water-heat transfer with airflow (Eqs. A.5, A.6 & A.7) | ACD-air-Snow | |
| SNW =0: No discrimination of snow and rainfall | SMETr=1: basic coupled water-heat transfer (Eqs. A.1 & A.2) | BCD-No-Snow | STEMMUS-FT_No-snow |
| | SMETr=2: advanced coupled water-heat transfer without airflow (Eqs. A.3 & A.4) | ACD-No-Snow | |
| | SMETr=3: advanced coupled water-heat transfer with airflow (Eqs. A.5, A.6 & A.7) | ACD-air-No-Snow | |

**Table 4. Comparative statistics values of various model versions for snow albedo, LE, soil temperature, and soil moisture. The best statistical performance is highlighted by bold fonts, while the values with poor statistical model performance is underlined with the italic fonts.**

| Experiments | | Statistics | Snow albedo | LE (mm/d) | Soil temperature (°C) | | | | | Soil moisture (cm³ cm⁻³) | | | | |
|---|---|---|---|---|---|---|---|---|---|---|---|---|---|---|
| | | | | | 5cm | 10cm | 20cm | 40cm | 80cm | 5cm | 10cm | 20cm | 40cm | 80cm |
| STEMMUS-FT_Snow | BCD | BIAS | -0.0100 | *0.162* | **-0.071** | *0.150* | -0.048 | -1.127 | -0.1390 | 0.0064 | 0.0091 | 0.0048 | **0.0031** | 1.80E-03 |
| | | R² | 0.296 | 0.278 | **0.976** | 0.958 | 0.881 | 0.626 | 0.810 | 0.704 | *0.586* | *0.310* | *0.387* | *0.237* |
| | | RMSE | 0.033 | 0.579 | 0.4697 | 0.415 | 0.544 | 1.548 | 0.5352 | 0.0194 | 0.0223 | 0.0307 | 0.0322 | 0.0118 |
| | ACD | BIAS | -0.0049 | -0.020 | -0.224 | **0.054** | -0.032 | -0.982 | **0.0129** | -0.0014 | **0.0024** | **0.0001** | 0.0045 | 7.57E-04 |
| | | R² | 0.253 | 0.232 | 0.964 | 0.969 | 0.971 | 0.944 | 0.995 | 0.878 | **0.960** | **0.991** | 0.992 | 0.982 |
| | | RMSE | 0.032 | 0.305 | 0.4462 | 0.374 | 0.209 | 1.190 | 0.1201 | 0.0087 | **0.0041** | 0.0028 | 0.0055 | 0.0019 |
| | ACD-air | BIAS | **-0.0048** | **-0.019** | -0.223 | 0.055 | -0.032 | -0.982 | 0.0130 | -0.0013 | 0.0025 | 0.0001 | 0.0045 | **7.55E-04** |
| | | R² | **0.338** | 0.217 | 0.963 | **0.969** | 0.971 | **0.944** | **0.995** | 0.883 | 0.960 | 0.990 | **0.992** | **0.982** |
| | | RMSE | **0.031** | 0.314 | 0.4464 | 0.374 | 0.210 | 1.190 | 0.1200 | 0.0084 | 0.0042 | **0.0028** | **0.0055** | **0.0019** |
| STEMMUS-FT_No-snow | BCD | BIAS | -0.0123 | *0.157* | -0.073 | *0.149* | -0.048 | -1.128 | -0.1397 | 0.0099 | 0.0092 | 0.0048 | 0.0031 | 1.70E-03 |
| | | R² | - | 0.303 | 0.976 | 0.958 | 0.881 | 0.627 | 0.810 | 0.771 | *0.581* | *0.309* | *0.386* | *0.240* |
| | | RMSE | 0.038 | 0.565 | 0.4673 | 0.415 | 0.544 | 1.548 | 0.5354 | 0.0261 | 0.0224 | 0.0307 | 0.0322 | 0.0117 |
| | ACD | BIAS | -0.0079 | -0.031 | -0.213 | 0.065 | -0.023 | -0.977 | 0.0154 | **-0.0010** | 0.0026 | 0.0002 | 0.0046 | 8.29E-04 |
| | | R² | - | **0.363** | 0.964 | 0.969 | **0.973** | 0.943 | 0.995 | **0.887** | 0.959 | 0.991 | 0.991 | 0.979 |
| | | RMSE | 0.037 | **0.242** | 0.4352 | **0.370** | **0.201** | 1.186 | 0.1210 | **0.0081** | 0.0044 | 0.0028 | 0.0058 | 0.0020 |
| | ACD-air | BIAS | -0.0079 | -0.031 | -0.210 | 0.072 | **-0.014** | **-0.968** | 0.0222 | -0.0011 | 0.0026 | 0.0003 | 0.0049 | 9.13E-04 |
| | | R² | - | 0.358 | 0.965 | 0.969 | 0.972 | 0.943 | 0.995 | 0.886 | 0.960 | 0.991 | 0.990 | 0.979 |
| | | RMSE | 0.037 | 0.243 | **0.4349** | 0.374 | 0.202 | **1.180** | **0.1198** | 0.0082 | 0.0041 | 0.0028 | 0.0061 | 0.0020 |

Note: $BIAS = \frac{\sum_{i=1}^{n}(y_i - \hat{y}_i)}{n}$, $R^2 = 1 - \frac{\sum_{i=1}^{n}(y_i - \hat{y}_i)^2}{\sum_{i=1}^{n}(y_i - \bar{y})^2}$, $RMSE = \sqrt{\frac{\sum_{i=1}^{n}(y_i - \hat{y}_i)^2}{n}}$, where $y_i$, $\hat{y}_i$, are the measured and model simulated values of the selected variable (snow albedo, LE, soil temperature/moisture); $\bar{y}$ is the mean values of the measurements of the selected variable (snow albedo, LE, soil temperature/moisture); $n$ is the number of data points.

The correlation is all significant at the 0.01 level, except for "-", which indicates that the correlation is not significant.

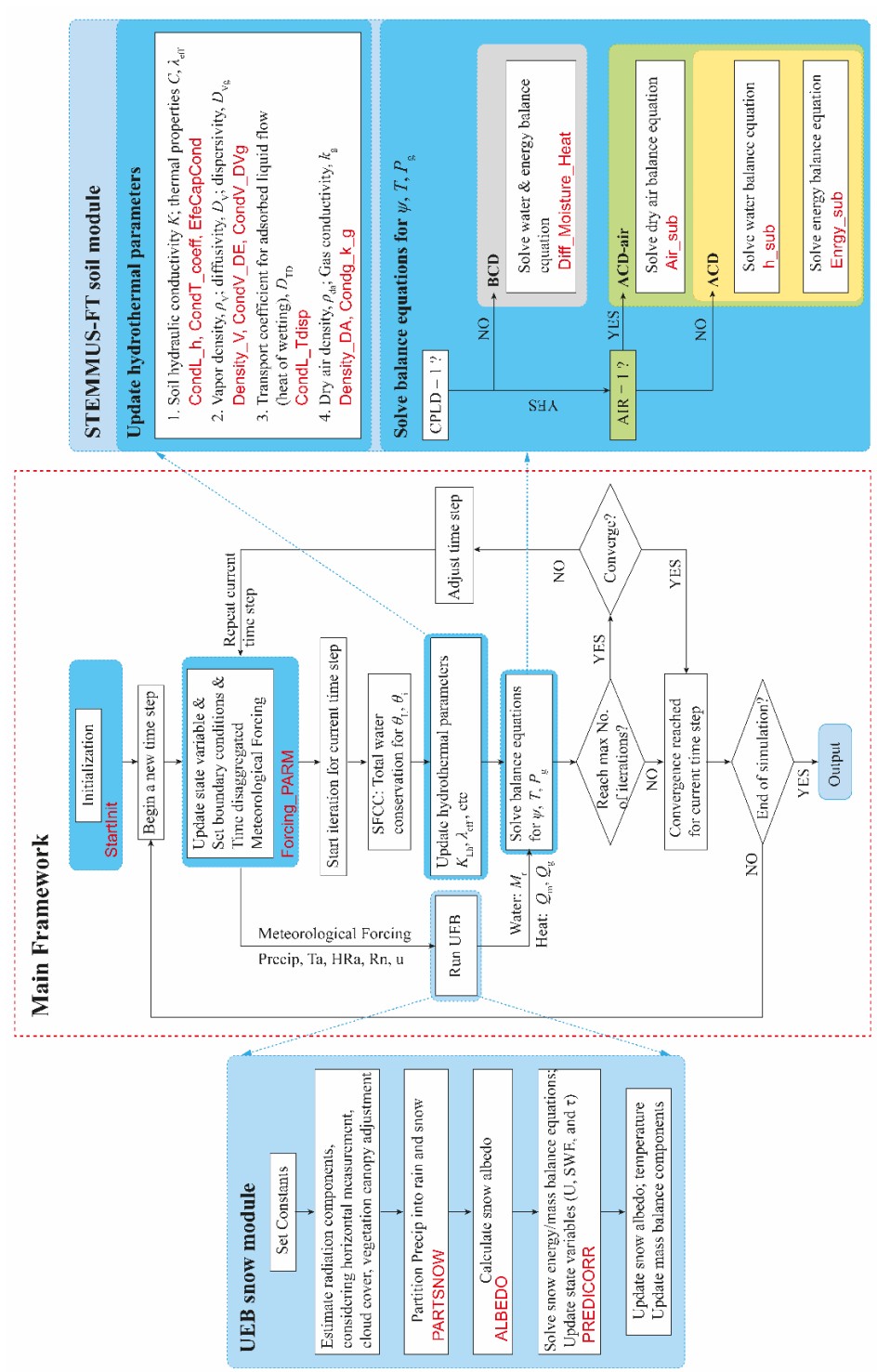

Figure 1. The overview of the coupled STEMMUS-FT and UEB model framework and model structure. SFCC is soil freezing characteristic curve; $\theta_L$ and $\theta_i$ are soil liquid water and ice content; $K_{Lh}$ is soil hydraulic conductivity; $\lambda_{eff}$ is thermal conductivity. $\psi, T, P_g$ are the state variables for soil module STEMMUS-FT (matric potential, temperature, and air pressure, respectively). U, SWE, and τ are the state variables for snow module UEB (snow energy content, snow water equivalent, and snow age, respectively). UEB, Utah Energy Balance module. Precip, Ta, HRa, Rn, and u are the meteorological inputs (precipitation, air temperature, relative humidity, radiation, and wind speed). $M_r$ is the snowmelt water flux, $Q_m$ is the convective heat flux due to snowmelt water and $Q_g$ is the heat conduction flux. Model subroutines are in red fonts.

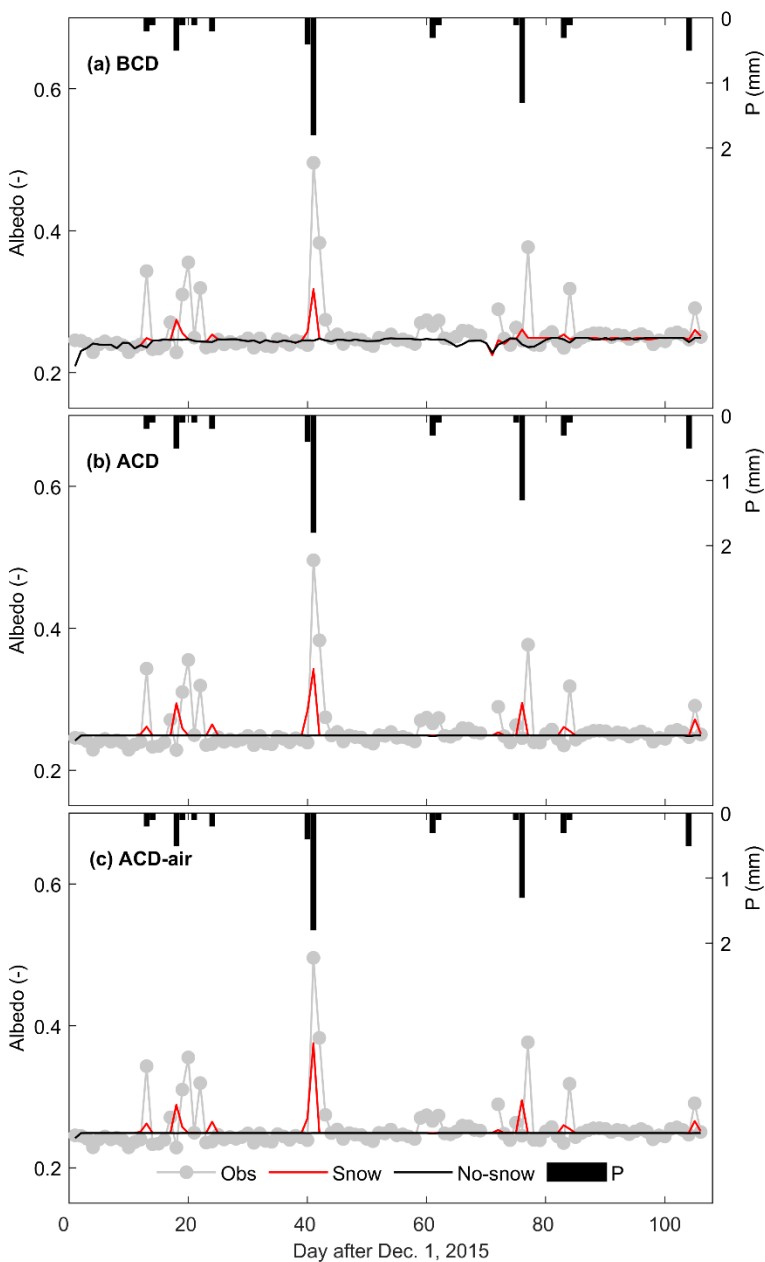

**Figure 2. Time series of observed and model simulated daily average albedo using (a) BCD, (b) ACD, and (c) ACD-air soil model with/without consideration of snow module, with the precipitation.**

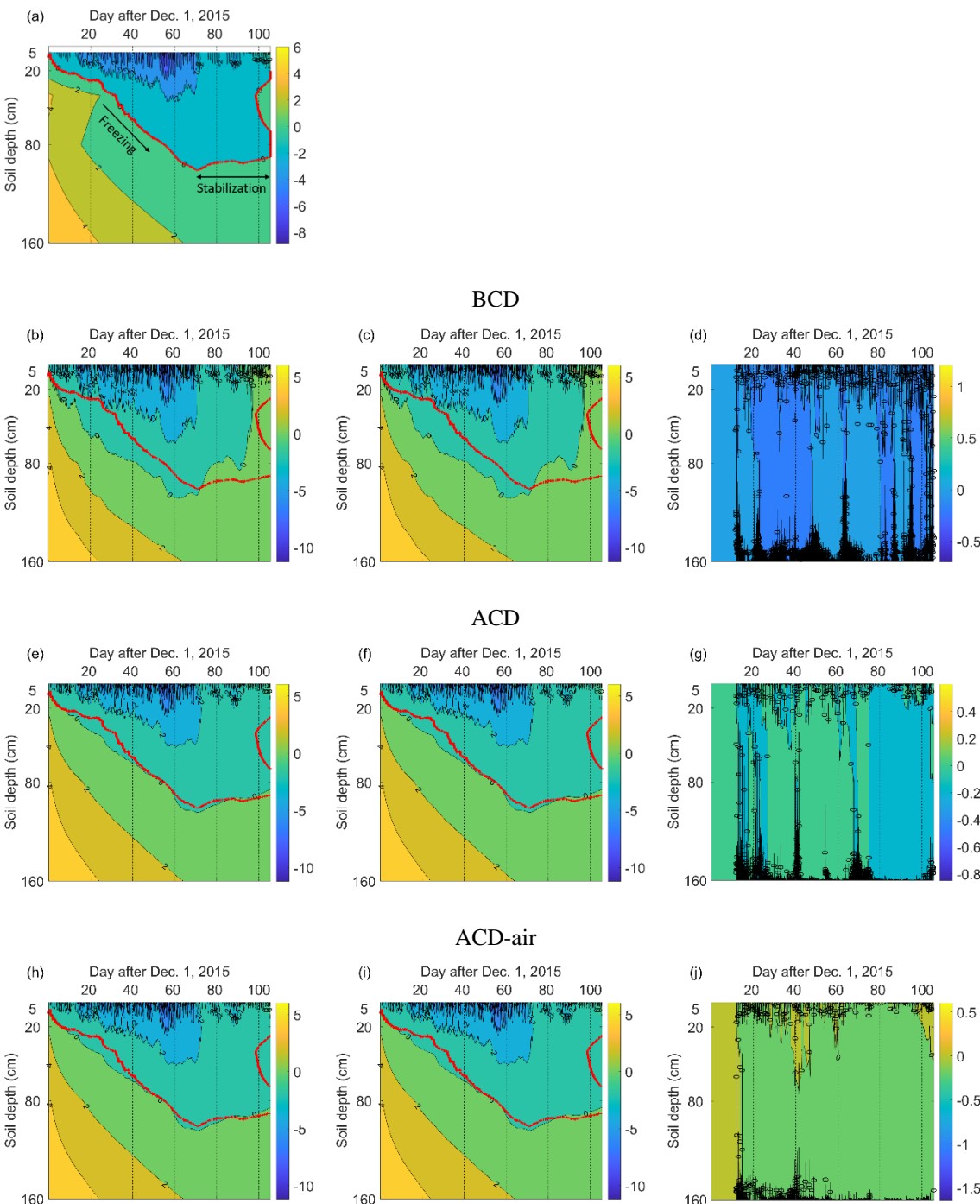

**Figure 3. The spatial and temporal dynamics of observed (a) and simulated soil temperature using BCD, ACD, and ACD-air soil model, with and without consideration of snow module (Snow: b, e, h and No-Snow: c, f, i) and the difference (d, g, j) (simulations with snow minus simulations without snow). The red line indicates the zero-degree isothermal line (ZDIL) from the measured soil temperature. The observed soil freezing stage and stabilization stage was marked in Fig. 3a.**

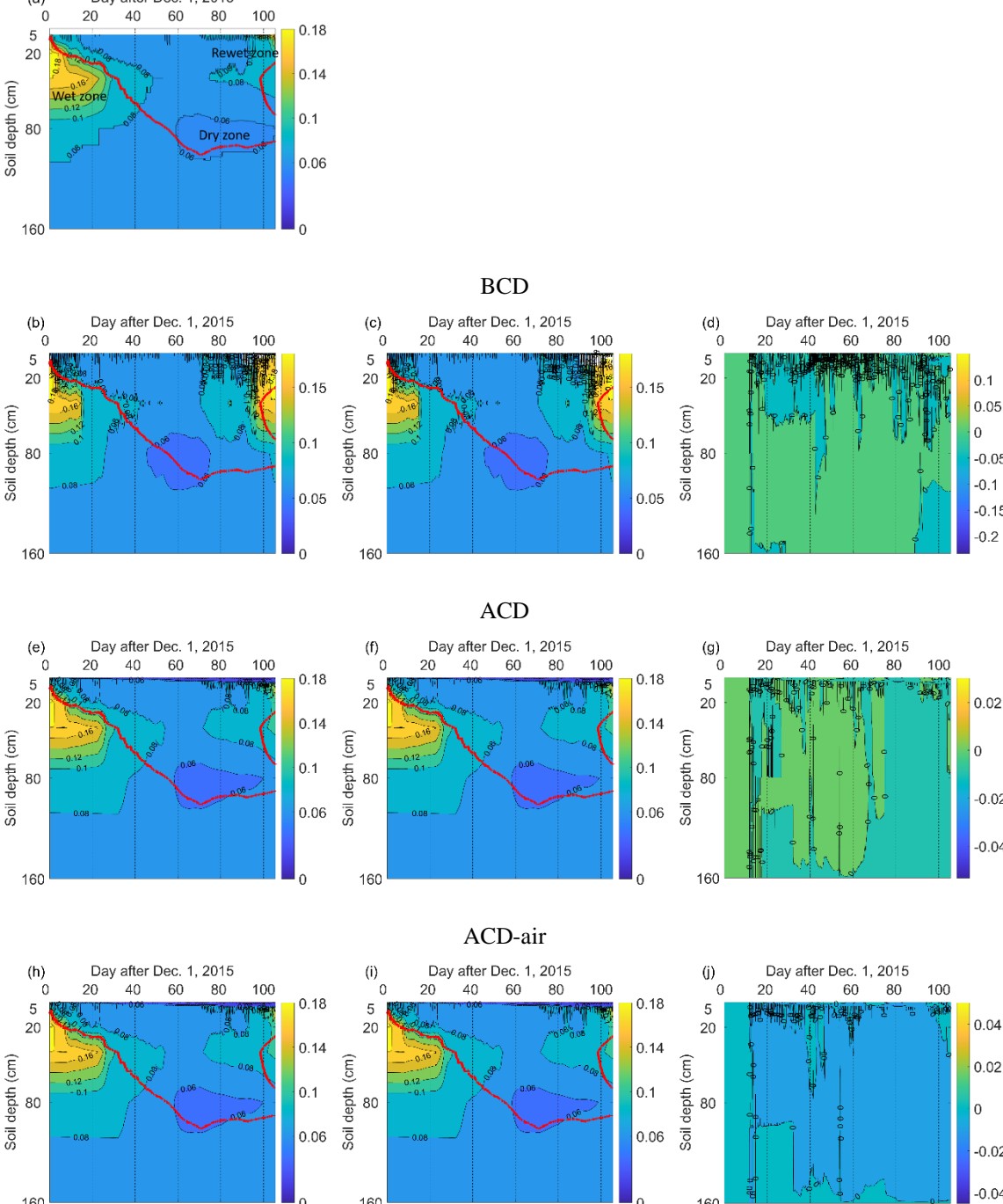

**Figure 4. The spatial and temporal dynamics of observed (a) and simulated soil volumetric water content using BCD, ACD, and ACD-air soil model, with and without consideration of snow module (Snow: b, e, h and No-Snow: c, f, i) and the difference (d, g, j) (simulations with snow minus simulations without snow). The red line indicates the zero-degree isothermal line from the measured soil temperature. The observed wet zone, dry zone and rewet zone of soil moisture was indicated in Fig. 4a.**

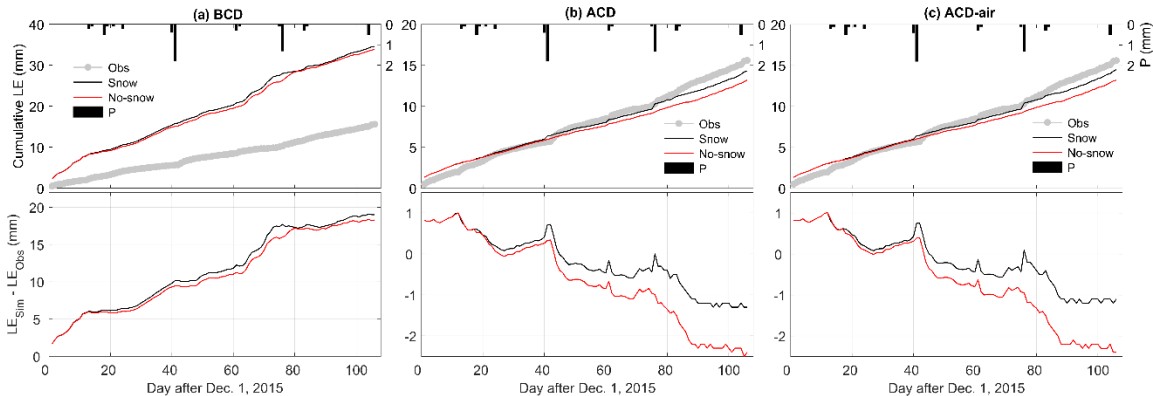

**Figure 5. Time series of observed and model simulated surface cumulative latent heat flux (LE) using (a) BCD, (b) ACD, and (c) ACD-air soil model with/without consideration of snow module, with the precipitation. The top row is the comparisons and the bottom row is the model bias of the cumulative surface LE.**

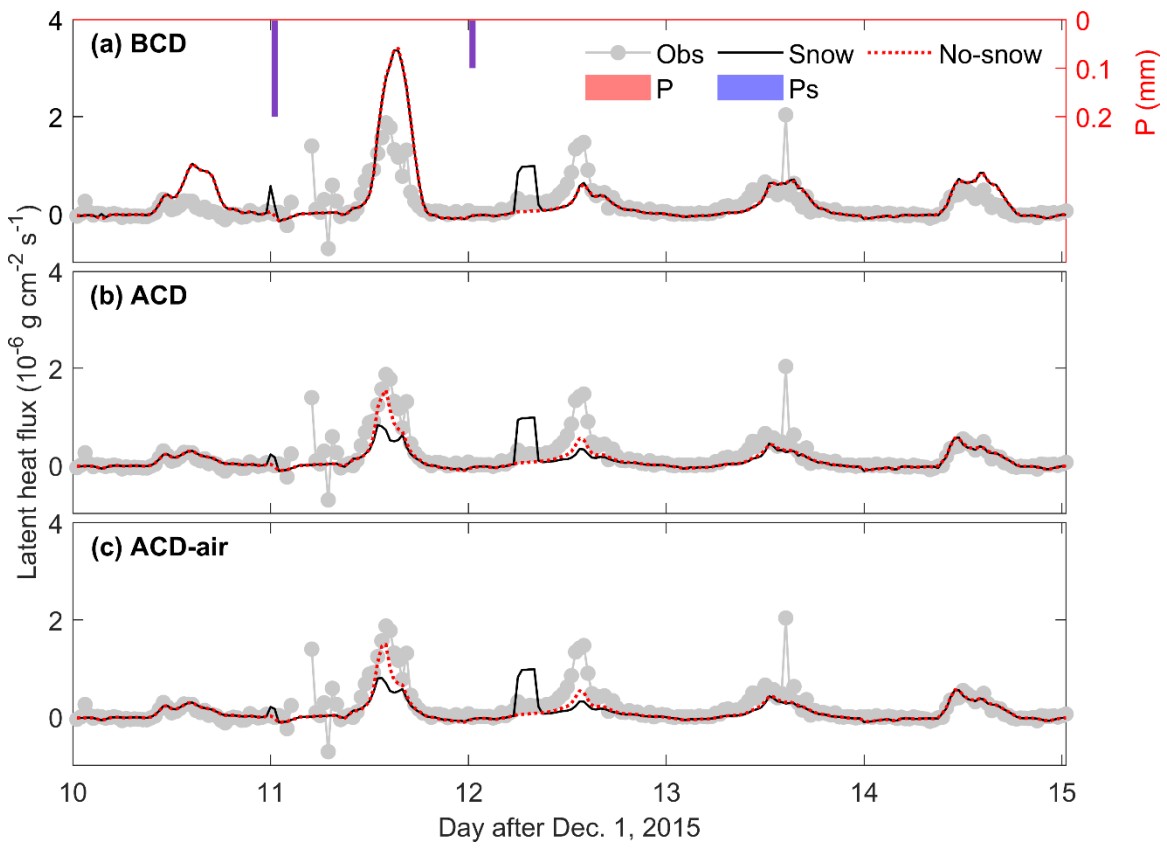

**Figure 6. Observed and model simulated latent heat flux, using (a) BCD, (b) ACD, and (c) ACD-air soil model with/without snow module, of a typical five-day freezing period (from 10th to 14th Days after Dec. 1. 2015). P is the precipitation and Ps is the snowfall. All precipitation is in the form of snowfall.**

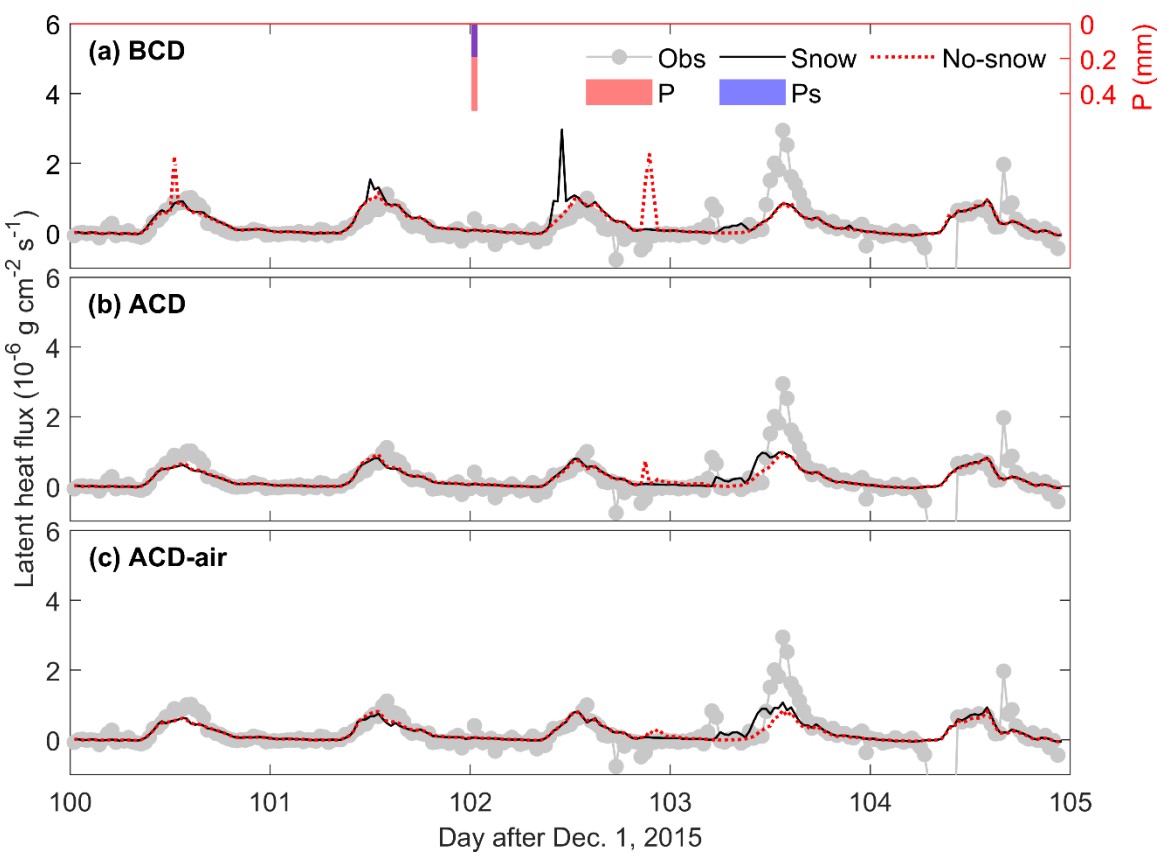

**Figure 7. Observed and model simulated latent heat flux, using (a) BCD, (b) ACD, and (c) ACD-air soil model with/without snow module, of a typical five-day thawing period (from 100th to 104th Days after Dec. 1. 2015). P is the precipitation and Ps is the snowfall.**

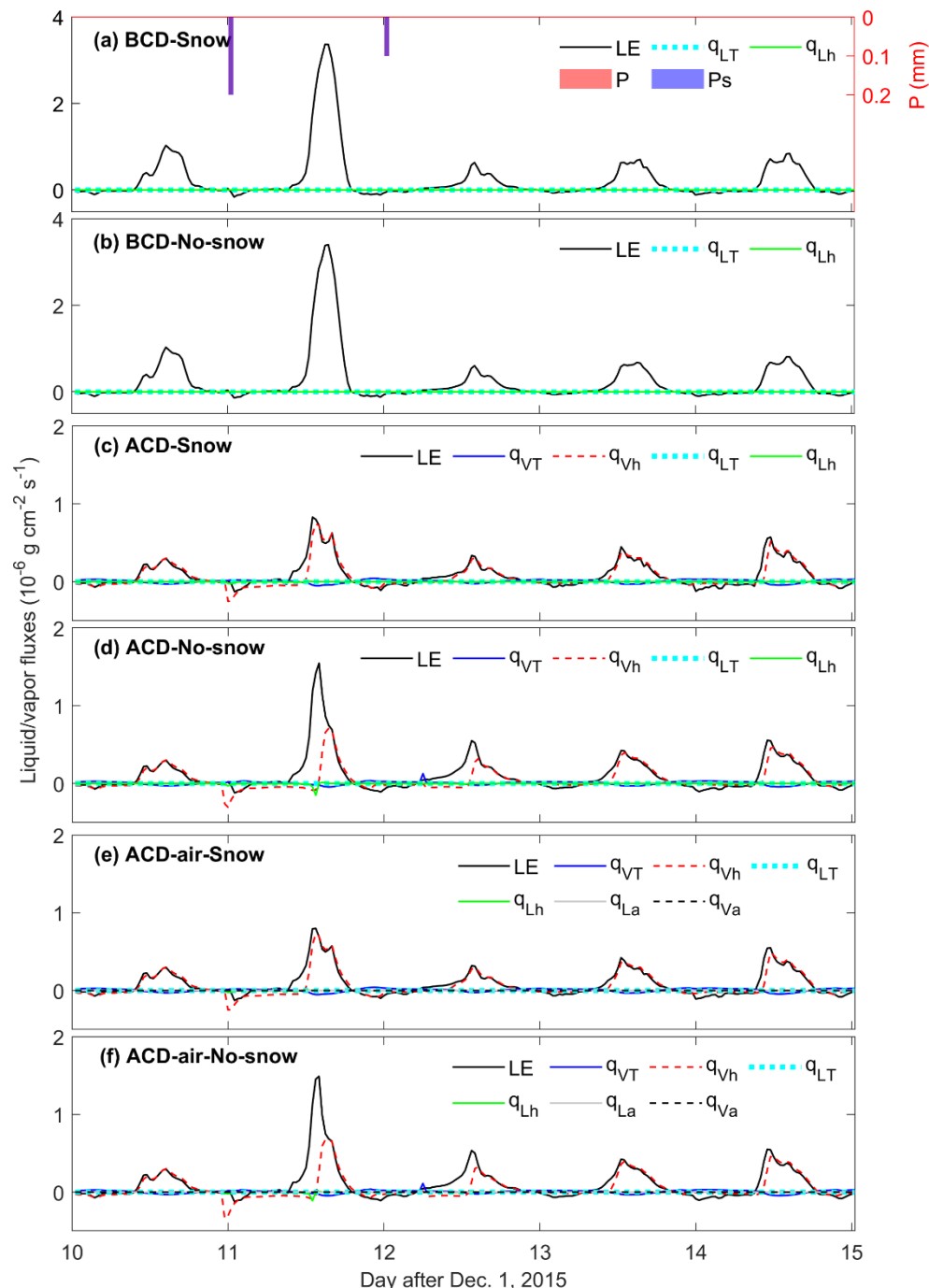

**Figure 8. Model simulated latent heat flux and surface soil (0.1cm) thermal and isothermal liquid water and vapor fluxes (LE, $q_{VT}$, $q_{Vh}$, $q_{LT}$, $q_{Lh}$, $q_{La}$, $q_{Va}$), with and without snow module, of a typical five-day freezing period (from 10th to 14th Days after Dec. 1. 2015). a, c, and e are the surface soil thermal/isothermal liquid water and vapor fluxes simulated by BCD-Snow, ACD-Snow, and ACD-air-Snow model, respectively. b, d, and f are the surface soil thermal/isothermal liquid water and vapor fluxes simulated by BCD-No-Snow, ACD-No-Snow, and ACD-air-No-Snow model, respectively. LE is the latent heat flux, $q_{VT}$, $q_{Vh}$ are the water vapor fluxes driven by temperature and matric potential gradients, $q_{LT}$, $q_{Lh}$ are the liquid water fluxes driven by temperature and matric potential gradients, $q_{La}$, $q_{Va}$ are the liquid and vapor water fluxes driven by air pressure gradients. Positive/negative values indicate upward/downward fluxes. Note that the surface LE fluxes without snow sublimation were presented. P is the precipitation and Ps is the snowfall. All precipitation is in the form of snowfall.**

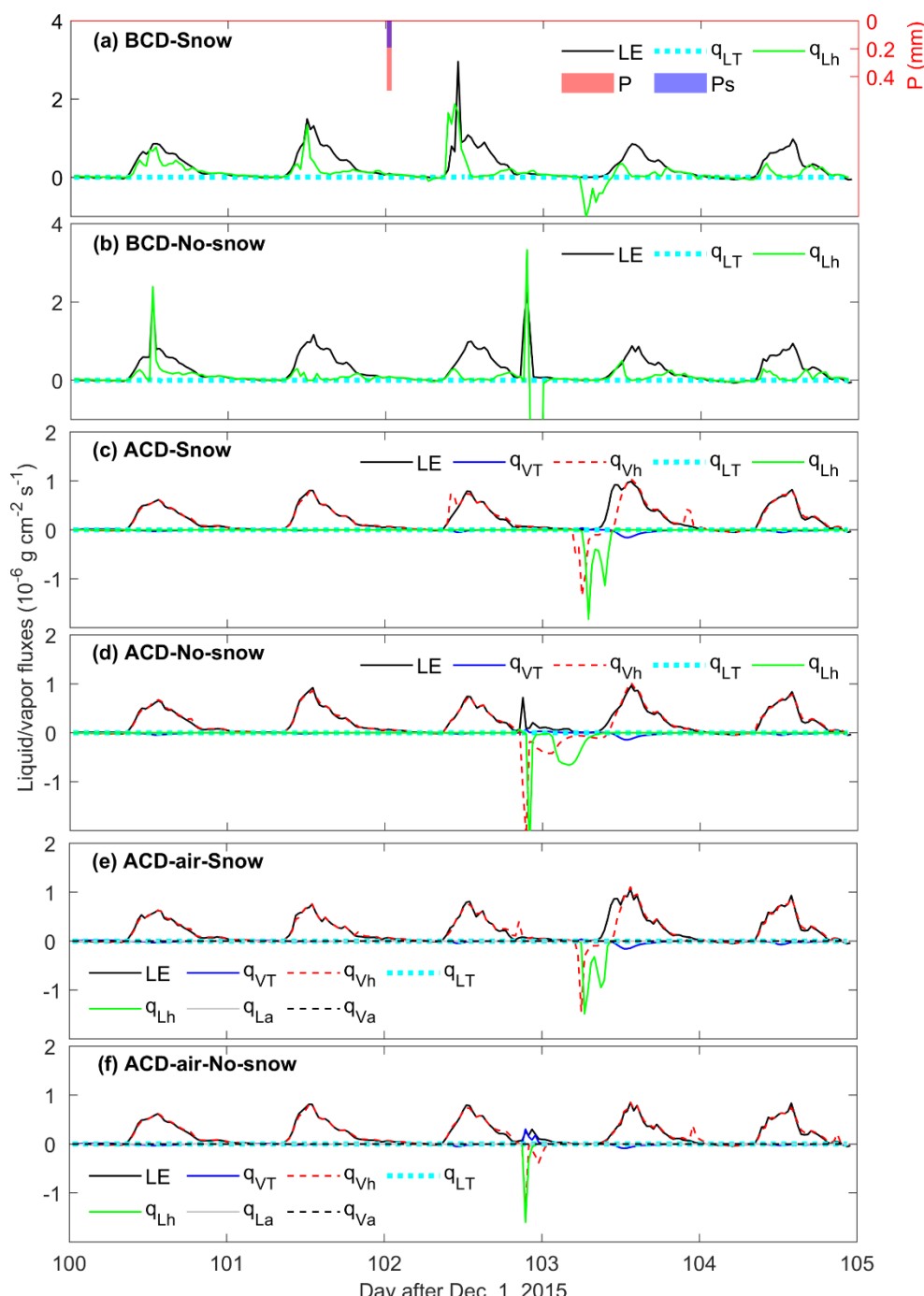

**Figure 9. Model simulated latent heat flux and surface soil (0.1cm) thermal and isothermal liquid water and vapor fluxes (LE, $q_{VT}$, $q_{Vh}$, $q_{LT}$, $q_{Lh}$, $q_{La}$, $q_{Va}$) using BCD (a, b), ACD (c, d), and ACD-air (e, f) simulations with and without snow module, respectively, during the typical 5-day thawing periods (from 100[th] to 104[th] Days after Dec. 1. 2015). a, c, and e are the surface soil thermal/isothermal liquid water and vapor fluxes simulated by BCD-Snow, ACD-Snow, and ACD-air-Snow model, respectively. b, d, and f are the surface soil thermal/isothermal liquid water and vapor fluxes simulated by BCD-No-Snow, ACD-No-Snow, and ACD-air-No-Snow model, respectively. LE is the latent heat flux, $q_{VT}$, $q_{Vh}$ are the water vapor fluxes driven by temperature and matric potential gradients, $q_{LT}$, $q_{Lh}$ are the liquid water fluxes driven by temperature and matric potential gradients, $q_{La}$, $q_{Va}$ are the liquid and vapor water fluxes driven by air pressure gradients. Positive/negative values indicate upward/downward fluxes. Note that the surface LE fluxes without snow sublimation were presented. P is the precipitation and Ps is the snowfall.**