# Peer review of "STEMMUS-UEB v1.0.0: Integrated Modelling of Snowpack and Soil Water and Energy Transfer with Three Complexity Levels of Soil Physical Process"

_Geoscientific Model Development, 2020_

## Author Response (AR1)

We thank the editor and reviewers very much for the dedicated time and efforts they put to improve/sharpen this manuscript with constructive comments. We made our point-by-point response in blue fonts as below. The referee comments are in black fonts.

**RC1**: Referee comments from #1 anonymous referee.

**RC2**: Referee comments from #2 anonymous referee.

Anonymous Referee #1

Referee comment on "STEMMUS-UEB v1.0.0: Integrated Modelling of Snowpack and Soil Mass and Energy Transfer with Three Levels of Soil Physical Process Complexities" by Lianyu Yu et al., Geosci. Model Dev. Discuss., https://doi.org/10.5194/gmd-2020-416-RC1, 2021

This manuscript aims to incorporate the snowpack effect into a STEMMUS-FT modeling framework, with various complexities of mass and energy transfer physics, then investigate the effect of snowpack on soil moisture and heat transfer. In general, the manuscript is well written and interesting to me. I recommend a major revision for this manuscript before its acceptance for publication.

We thank the reviewer very much for the time and effort and also for the insightful comments. Please see our specific response in blue fonts below.

**General comments:**

**[1] RC1:** There are too many long sentences which make them hard to follow.

**Response:** We have made changes, separate the long sentences into short ones, trying to make it easy to read and follow.

**[2] RC1:** Can the simulated time series of daily average albedo and LE (latent heat flux) be longer?

**Response:** We thank the reviewer for the insightful comments. Yes, the temporal spectrum of daily average albedo and LE (latent heat flux) can be longer. For instance, we have run about 3-year simulation (Mar. 2016 – Aug. 2018) of surface energy fluxes, including radiative components, sensible heat fluxes and LE, against the in-situ observations for this site. It well demonstrated the performance of STEMMUS-FT (Yu et al., 2020). The UEB model also shows its widely and successful application spanning a variety of hydrological conditions (Table S3). All these give us confidence that the integrated STEMMUS-UEB model can be applicable to this experimental site. On the other hand, we have to admit that as the harsh environment, the dataset is difficult to achieve and to have it fully corroborated. The accuracy of precipitation measurement (both the amount and its liquid/solid fractions), which is important to have the snowpack dynamics right, is of uncertain and needs effort to have more dataset to constrain it. We add some text in Section 4.1 Limitations (Line 342).

For this work, we focused on developing the integrated STEMMUS-UEB model (in Sect. 2 and Supplement). Furthermore, upon the confirmation of STEMMUS-UEB model performance, we are trying to emphasize/demonstrate its capability for understanding the effect of snowpack on the subsurface soil water and heat transfer processes. With the aid of both in-situ measurements and numerical experiments, we can see that: i) the presence of snowpack can be identified by the abrupt dynamics of daily average albedo. STEMMUS-UEB model well captured the large abrupt of daily average albedo with the precipitation. ii) models considering the snowpack process generally presented better simulation performance than models without snowpack. Then we further illustrated the capability of STEMMUS-UEB, in terms of understanding how snow water infiltrates downwards and interacts with subsurface soil water and heat regimes.

**[3] RC1:** The freezing and melting processes are a cyclic process, it will be more reasonable to describe the two processes together (section3.4.1 & 3.4.2).

**Response:** We agree that freezing and melting processes are inherently bounded together. We made changes about section 3.4.1 & 3.4.2 according to reviewer's comments. First, we presented diurnal dynamics of the observed and model simulated latent heat flux (LE) during the rapid freezing and thawing periods with precipitation events in Figure 6 and 7, respectively.

Then the relative contribution of individual flux components to the total mass transfer were presented in Figure 8 and 9, respectively, for the freezing and thawing periods.

The freezing and thawing processes were described together in Section 3.4.

**[4] RC1:** The language could be polished in various places in order to facilitate understanding.

**Response:** Thanks a lot for the helpful comments. We had the manuscript English edited.

**Specific comments:**

**[5] RC1:** The overview of the coupled STEMMUS-FT and UEB model framework and model structure in figure 1, the text is too small to read.

**Response:** We made modifications of the text in figure 1 and also made it as Landscape orientation instead of Portrait orientation.

**[6] RC1:** Figure 6 is too long, which can be divided into three figures or rearranged. Time series of different variables overlapped and changes in different variables are not visible. Such as qLh and qLT. Same as other figures.

**Response:** Figures were rearranged to make it easy to read. Figure 6 and Figure 7 were further presented as Figure 6, Figure 7, Figure 8, and Figure 9. The state variables were presented together as a cyclic freezing-thawing process. The comparison of observed and model simulated latent heat fluxes (LE) for the freezing and thawing periods were given sequentially in Figure 6 and Figure 7. Figure 8 and Figure 9 present the model simulated latent heat flux and surface soil thermal and isothermal liquid water and vapor fluxes for the freezing and thawing periods (Section 3.4).

We use different type of lines (dotted line for $q_{LT}$, dashed lines for $q_{Vh}$, and $q_{Va}$, and solid lines for $q_{VT}$, $q_{Lh}$, $q_{La}$, with different colors) to make the different flux component visible in figures.

**[7] RC1:** In Figure 7 (e, f, h, i), the sharp changes should be explained on Day 103. The figure and legend are overlapped.

**Response:** We moved the legend a little bit to avoid its overlapping with the figure content. For the sharp changes of isothermal liquid and vapor fluxes ($q_{Lh}$, $q_{Vh}$), it is due to the large increase of surface soil moisture after the precipitation events (see Supplement Figure S2 d & f). This resulted in the large gradient of matric potential thus the sharp changes of isothermal liquid and vapor fluxes. We add the explanation in Section 3.4 (Line 321).

Anonymous Referee #2

Referee comment on "STEMMUS-UEB v1.0.0: Integrated Modelling of Snowpack and Soil Mass and Energy Transfer with Three Levels of Soil Physical Process Complexities" by Lianyu Yu et al., Geosci. Model Dev. Discuss., https://doi.org/10.5194/gmd-2020-416-RC2, 2021

**[1] RC2:** This study presents a new integrated modelling of snowpack and soil water/energy transfers, called STEMMUS-UEB, presenting three levels of soil transfer complexities. The model is evaluated on one site equipped with soil temperature, moisture and energy fluxes sensors. The performances of the 3 options of the model are discussed. This is an interesting paper but quite difficult to follow and some questions need to be addressed before further consideration for publication.

**Response:** We thank very much the reviewer for the insightful comments. Please see our specific response as below.

**[2] RC2:** A general issue is that the test site seems to be poorly influenced by snow. I am therefore wondering if it is really appropriate for the model evaluation.

**Response:** Great thanks for this critical comment. As partly explained in **[2] RC1**, this work is to describe the integrated soil-snow model STEMMUS-UEB and further confirm that STEMMUS-UEB model can identify and understand the snowpack effect, even for regions with the intermittent snowfall events.

The selected site is covered by a seasonal frozen ground with mostly episodic snowfall events. Compared to the sites with heavy snow events and thick snow layer, this site is indeed less influenced in some sense, e.g., snowmelt runoff. Nevertheless, the snow accumulation and snowmelt infiltration and its effects on the subsurface soil can still be

identified, which indicates how sensible the STEMMUS-UEB can represent such intermittent snowfall events.

In addition, such conditions (seasonal frozen ground with episodic snowfall events) commonly exist in the high mountain cold regions around the globe, while its implications for subsurface water and heat dynamics (therefore, the subsequent impact on land-atmosphere interactions) are rarely studied. Nevertheless, the STEMMUS-UEB can be applied for the sites with thick snow cover or perennial snowpack, since the coupled model is constructed to account for the physically-based processes.

From the perspective of experimental measurements, it is indeed helpful to enrich the snowpack-relevant data (e.g., high resolution observation of the spatiotemporal field of wind speed, precipitation, and snowpack variations) and make them more constraint and less of uncertainty. We add some text in Section 4.1 Limitations (Line 342).

**[3] RC2:** The model description in the main paper lacks on the description of the thawing/freezing processes: how is the fraction of liquid/solid water calculated and what about the soil hydraulic conductivity? how the rainfall/snowfall partition is done?

**Response:** We added the description of thawing/freezing process in Section 2.1 (Line 98 and Line 124).

a. how is the fraction of liquid/solid water calculated and what about the soil hydraulic conductivity?

The frozen soil physics considered in STEMMUS-FT include three parts: i) the ice blocking effect on soil hydraulic conductivities (see Supplement Sect. 2.2.2); ii) the inclusion of ice effect in the calculation of soil thermal capacity/conductivity (see Supplement Sect. 2.2.8); iii) the exchange of latent heat flux during phase change periods. With the aid of Clausius Clapeyron relation, which characterizes the phase transition between liquid and solid phase in the thermal equilibrium system. The soil water characteristic curve (e.g., van Genuchten, 1980) is then extended to consider the

freezing temperature dependence, i.e., soil freezing characteristic curve (Hansson et al., 2004; Dall'Amico et al. 2011). The fraction of soil liquid/solid water at a given temperature was then calculated prognostically with the soil freezing characteristic curve. Soil hydraulic parameters were further used in the Mualem (1976) model to compute the soil hydraulic conductivity. The ice effect is considered by reducing the soil saturated hydraulic conductivity as the function of ice content (Yu et al., 2018).

b. how the rainfall/snowfall partition is done?

Two precipitation types, i.e., rainfall and snowfall, are discriminated by the dependence on air temperature.

[4] RC2: Figure 2 is too small and difficult to read

Response: We rescaled the y-axis (from [0,1] to [0.1,0.7]) and enlarged Figure 2 to make it easy to read.

[5] RC2: Figures 6 and 7 are also difficult to understand: the precipitation events are rainfall or snowfall? what is the amount of SWE during that periods? It is surprising to see that the model without snow modeling performs generally better in the simulation of the latent heat flux compared to the snow model. It would be necessary to elaborate a bit more on that result.

Response: Figure 6 and 7 were modified correspondingly. Both the precipitation amount and the simulated snowfall (SWE) component were presented in the updated figures. The rainfall is the precipitation minus snowfall component.

For the discrepancies in terms of latent heat flux for the selected periods, it is possibly due to the inaccurate precipitation measurements and interpretation, in terms of either the amount, time of precipitation, or the partition of precipitation into solid and liquid forms. Moreover, the simple air temperature-precipitation type relation maybe not

suitable for this region. As argued by Ding et al. (2014), air temperature is not the best indicator of precipitation types. Other factors, i.e., relative humidity, elevation, and wet-bulb temperature, are also very relevant and should be taken into account. The uncertainties in discriminating the precipitation types can be the possible reason here. The episodic snowfall events are challenging to be well captured and simulated by the current snowpack models. The snowpack accumulation, melting process and water and energy partitioning of snowpack into snow sublimation and the snowmelt are with uncertainties as well.

We add some more text to explain and discuss such limitations in Section 4.1 (Line 349).

Nevertheless, this work focused on identifying the snowpack impacts on the underlying subsurface water and heat dynamics. And, the difference between the model simulations with and without the snow module can be attributed to the different surface water and energy regimes. Models without snow regards the precipitation as the rainfall, i.e., liquid form of water, adds on the topsoil surface immediately. Most of the incoming water directly contributes to the infiltration process.

While for the model with snow module, it considers snowpack related processes, accumulation, sublimation, melting process and then the water infiltration process. Compared with the model without snow module, the increase of surface soil moisture is usually delayed and less significant.

The difference in surface water status results in different gradients of matric potential. Models without snow have larger gradient of matric potential for the top surface soil layer. Then more amount of isothermal liquid and vapor fluxes ($q_{Lh}$, $q_{Vh}$) were generated contributing to the total latent heat flux.

Only with consideration of two-phase flow (ACD, ACD-air), the difference between models with and without snow module can be identified during the daytime after winter precipitation events. Generally, from the foregoing, considering the vapor flow/airflow retarded the total surface water transfer (Figure 6 a vs. b/c).

Identifying the difference and understanding what lies behind these differences between models with and without snowpack can be only made by two phase flow model (ACD, ACD-air). These are the highlighting points and benefits of the developed integrated soil-snow model STEMMUS-UEB.

**[6] RC2:** In the title, I suggest to replace "mass" by "water" to be more precise.

**Response:** We replace "mass" with "water" in the title.

**[7] RC2:** The English need to be revised

**Response:** We carefully revised the English as suggested.

**[8] RC2:** The abstract need to be rewritten to better highlight the main findings of the work

**Response:** We rewritten the abstract to highlight the main take home messages of this work.

The main findings of the work are briefly summarized here as:

i) we developed an integrated soil-snow-atmosphere model, STEMMUS-UEB, which takes advantage of the easily transferable and physically based description of the snowpack process by UEB snowmelt model and the detailed interpretation of the soil physical process by STEMMUS-FT model.

ii) the proposed model can well capture the dynamics of daily average albedo, latent heat flux, and the snowpack effect.

iii) three mechanisms, i.e., surface ice sublimation, snow sublimation and increased soil moisture, contribute to the enhanced latent heat flux after winter precipitation events.

iv) Physically realistic analysis of the snowpack effects (e.g., LE enhancement) can only be reproduced by the advanced coupled STEMMUS-UEB (ACD, ACD-air). The basic coupled version of STEMMUS-UEB (BCD) models, however, cannot provide a realistic description of the soil water and heat transfer.

**Reference**

[revised manuscript text omitted]

---

## Author Response (AR2)

This paper evaluates the simulation results on the liquid-vapor-air flow mechanisms between snowpack and soil by coupling a snow module and a soil module. The authors made great efforts to couple the models and made a detail analysis of their results. However, three major concerns are rising from the unclear purpose of this article, the weakness of the coupling method, and the unconfident results due to missing snow observation.

We thank the reviewer very much for the time processing our manuscript. We made our point-by-point response as below.

Major concerns:

1. What do the authors want to find by considering various complexities of mass and energy transfer physics? Isn't a more comprehensive parameterization scheme, without any simplification, is better for a snow-soil model? Are there any physically-based conflicts between different sub-models? Or are any sub-models a fault, so the authors had to find it out by this coupling?

Response: From the model process understanding perspective, we agree that a more comprehensive parameterization scheme is a better choice. From the application perspective, most of current land surface models or snow-soil models usually adopt a simplification of soil mass and energy transfer scheme (basic water and heat coupled, with no vapor flow and airflow). It has been reported that the consideration of the tightly coupled water and heat transfer scheme and vapor flow and airflow is important for the realistic interpretation of the soil hydrothermal dynamics in cold regions. However, the relative role of water and heat transfer scheme, vapor flow, airflow in portraying the soil hydrothermal regimes with snowpack is rarely presented. This work is trying to figure out whether the difference in soil physical processes affects the simulations and what is the role of individual processes in the soil hydrothermal state simulations with the snowpack. It helps to find under what conditions the specific processes are important and cannot be overlooked. Thus it can provide the guidance on the model development about how complex the soil physical processes should be taken into account in cold regions.

We added some relevant text in the Introduction part. See Line 76-91.

2. The key of this paper is the heat-water interaction between UEB and STEMMUS-FT. However, the UEB model may not be a good choice since it considers the snowpack as a 1-layers object, ignoring the significant temperature change on different snow profiles. The authors need to clarify: 1) whether the UEB model gives the temperature of snowpack bottom since it is the most critical boundary condition for solving soil heat transfer. 2) whether the water flows from snow would bring heat to soil. If YES, how did you considered the water flow temperature? 3)whether the solar radiation penetration was considered in your method since the snowpack would be very shallow in the Tibetan Plateau.

Response: Thanks a lot for the critical comments. Using the developed model, this manuscript is currently focus on investigating the effect of snowpack on soil hydrothermal dynamics. UEB model is selected for its simple yet physically based snowpack energy balance parameterization,

which can well simulate the snowpack dynamics. We agree that the temperature profile within the snowpack is not captured well by the single layer snowpack model UEB (UEB only explicitly simulates the snow surface temperature and the average snowpack temperature). The temperature of snowpack bottom is not given by the UEB model. Here we used the in situ surface temperature measurements as the boundary condition for solving the soil heat transfer, by which the thermal effect of snowpack was implicitly considered. For our next step, we are willing to implement the energy balance as the boundary condition for the soil heat transfer. As long as the snowpack is presented, the soil surface energy balance was recalculated to consider the effect of the snowpack on the energy budgets. It includes explicitly the convective heat due to the melt snow water, conductive heat, and the solar radiation attenuation.

In addition, the radiation components were also affected as the presence of the snowpack. In UEB model, when the snowpack is shallow, the albedo is weighted between the snow albedo and bare ground albedo $A_{v/ir} = r\alpha_{g,v/ir} + (1 - r)\alpha_{v/ir}$. The weighting factor $r = \left(1 - \frac{z}{h}\right)e^{-z/2h}$, in which the solar radiation penetration in the snow is exponentially attenuated as expressed by the exponential term. The incoming radiation is then altered, and the other energy budget components will be affected accordingly.

We added the relevant clarification in the Section 2.1 and 2.3.

3. The topic is the coupling between parameterizations of snow and soil. However, there is no direct snow observation such as snow depth. The ALBEDO or other variables could only provide indirect evidence. So, it is not confident for me to agree with the authors on the benefits of STEMMUS-UEB. I suggest the authors using a more comprehensive dataset to evaluate your model. Some supersites for the cold region, even in Tibetan Plateau, such as the super snow and frozen-ground observations network in Qilian mountain (Che et al.,2019), could provide enough dataset for your evaluation and similar job (Li et al., 2019).

Che, T., Li, X., Liu, S., Li, H., Xu, Z., Tan, J., Zhang, Y., Ren, Z., Xiao, L., Deng, J., Jin, R., Ma, M., Wang, J., and Yang, X.: Integrated hydrometeorological, snow and frozen-ground observations in the alpine region of the Heihe River Basin, China, Earth Syst. Sci. Data, 11, 1483–1499, https://doi.org/10.5194/essd-11-1483-2019, 2019.

Li, H., Li, X., Yang, D., Wang, J.,Gao, B., Pan, X., et al. (2019).Tracing snowmelt paths in an integrated hydrological model for understanding seasonal snowmelt contribution at basin scale. Journal of Geophysical Research: Atmospheres,124.

**Response:** We thank the reviewer very much for the comments.

Albedo has been demonstrated significantly correlated to the snow cover. This is also true for the super snow station Yakou in Qilian mountain (Figure R1). Together with the auxiliary meteorological data, the dynamics of surface albedo can be used to properly infer the presence of snowpack and the period during which it affects land surface processes (please see Table R1, Figure R1 and Figure R2, using Yakou super snow station data as an example). Given that results, we can take the albedo as the valid indicator of the presence and influencing period of the

snowpack. As such, we conducted the analysis of the effect of snowpack on the soil hydrothermal regimes.

We used the dataset suggested by the reviewer and ran a simulation using the developed model. The results were presented in the Appendix B. The simulation results demonstrated the validity of the STEMMUS-UEB model. The difference between models with and without the snow module is clearly presented in Figure B1-B5.

Minor comments:

4. Line 21-22. It may not be suitable to use "the effect of snowpack on soil moisture and heat transfer" since there is the only comparison among different parametrization schemes, but not a reliable analysis of this effect based on the examined model and its proven results.

**Response:** Following the relation between surface albedo and snowpack, we can confirm the presence of the snowpack and its lasting time with the meteorological data. It is also validated from the Yakou super snow station in Qilian mountain. We added the figures indicating the relationship between the surface albedo and snowpack. Table R1 and Figure R1-R2 show that the surface albedo can be a reliable indicator for the presence of snowpack.

With and without the snow module, the precipitation and following processes are treated differently. Only with the snow module, the precipitation can be partitioned into the rainfall and snowfall components. The snow accumulation and melt processes are explicitly considered. The water reaching the soil surface is thus altered both in the amount and time. Given the different incoming water flux, we investigate how the models with different soil physical processes respond and further connect it to the observed state variables.

5. Line 69-70. Many models such as SHAW, CLM, and CROCUS all consider the underlying soil physical processes. In addition, what is the meaning of air balance in Table 1? If the air balance has notable influences on the modeling, please explain it with some references.

**Response:** We agree that many models consider both the snowpack and the underlying soil physical processes. Nevertheless, most of them do not implement the multi-parameterization of the soil physical processes and rarely consider the role of airflow.

Air balance represents the conservation of air fluxes in soil. Dry air transfer in the soil is driven by the dry air concentration/density gradient and the air pressure gradient. It includes four parts: 1) the diffusive flux (Fick's law, $D_e \frac{\partial \rho_{da}}{\partial z}$), driven by dry air density gradient; 2) the advective flux (Darcy's law, $\rho_{da} \frac{S_a K_g}{\mu_a} \frac{\partial P_g}{\partial z}$), driven by the air pressure gradient; 3) the dispersive flux (Fick's law, $(\theta_a D_{Vg}) \frac{\partial \rho_{da}}{\partial z}$); and 4) the advective flux due to the dissolved air (Henry's law, $H_c \rho_{da} \frac{q_L}{\rho_L}$).

The effects of airflow on soil mass and energy transfer are two-fold. First, it directly results in the additional water and vapor fluxes and corresponding convective heat flow. Second, the presence of airflow in soil pores alters the water vapor status (density, pressure) thus the vapor transfer processes, which have been demonstrated important in soil water and heat transfer.

Several papers have reported the influence of airflow on the soil water and heat transfer. It can significantly retard the infiltration (Touma and Vauclin, 1986; Prunty and Bell, 2007), enhance the evaporation after precipitation (Zeng et al., 2011a, b; Zeng and Su, 2013), and cause the convective heat transfer and thus the temperature difference between the upper and the lower part of a permafrost talus slope (Wicky and Hauck, 2017).

We added the relevant description in the Introduction part of the updated manuscript. See Line 76-91.

6. Section 2.1 is just like an introduction of the SEMMUS model, but the readers would be firstly interested in your methods in this paper. I suggest presenting the coupling method first and then detail the two models separately.

**Response:** We adjusted this section and presented the coupling method first and then detail the two models as further explanation. A paragraph summarizing this section was also presented.

Changes in the manuscript are:

"This section first presents the coupling procedure of STEMMUS-FT and UEB model, followed by the detailed description of the two models and their successful applications. Then the used model configurations and two tested experimental sites in the Tibetan Plateau were elaborated. Maqu case is for investigating the effect of snowpack on the underlying soil hydrothermal regimes. Yakou case is for demonstrating the validity of the developed STEMMUS-UEB model in reproducing the snowpack dynamics (results were presented in Appendix B)."

7. Line 94. What is the job this reference cited?

**Response:** Clark et al. (2015) advocated that it is necessary to develop the integrated model, under the same model structure, with various parameterization of land surface processes. In such way, it avoids the misinterpretation of the model-comparison results and focus on the comparison of the relevant processes.

We deleted this reference here.

8. Line 100. It may not be the only linkage because the water flow would transfer heat and mass simultaneously.

**Response:** Yes. Many thanks. We agree that water flow can transfer the mass and heat flow. The convective heat flux transferred by the water flow is usually less than that by the vapor flow.

We rephrased the sentence as "the primary linkage between soil water and heat flow". Line166.

9. Line 117. Why first order? A little weird.

**Response:** We use the first order to stress that the UEB model can capture the primary characteristics of snowpack well. We deleted the "first order".

10. Line 127. Could you please specify the means of the one-way?

**Response:** Currently, as we more focus on investigating the influence of snowpack on subsoil hydrothermal regimes, the feedback effect of soil hydrothermal dynamics on the snow dynamics was relaxed. The surface temperature measurements were considered as the topsoil energy balance boundary conditions. In such way, the topsoil boundary condition is reliable, and the interactive soil-snow effect is implicitly considered.

From the perspective of water fluxes, the meltwater from snowpack was added to the subsurface soil, in addition to the rainfall, as the topsoil boundary condition for solving soil water transfer.

We focus on investigating the effect of the snowpack on the subsurface soil water and vapor transfer. And, thanks for your comments, we found the 'one-way' somewhat confusing and deleted it. In addition, we added some explanation text in the updated Section 2.1 accordingly.

11. Line 132. Did the UEB model simulates the meltwater temperature and includes it in the heat flux output? I am also interested in the temperature of the snowpack bottom that the UEB model could give or not. As the authors suggest, the UEB model assumed the snowpack as a single layer, so the snowpack temperature is very different from the underlying boundary temperature of the snowpack, which is the most crucial heat boundary for solving the heat transfer in the soil matrix. Also, solar radiation should be considered since it could penetrate about ~10 cm into the snow if there are shallow snowpacks.

**Response:** We agree with the reviewer that the effect of snowpack on energy budgets are important. Currently, UEB can provide the snow heat content, which is for estimating the snowpack average temperature. The convective heat flux due to the meltwater can be simulated by UEB as the heat flux output. UEB cannot provide the temperature of the snowpack bottom for the heat boundary condition solving the soil heat transfer. The current solution is to take the *in situ* surface temperature as the topsoil heat boundary. The aim of this work is first to identify the presence of snowpack and then investigate the effect of snowpack on the subsurface soil water and vapor transfer processes.

In the following updates, your suggestions were to be implemented. The energy balance conservation will be set as the topsoil heat boundary condition. The snowpack meltwater heat flux, attenuation of solar radiation by the shallow snowpack and albedo changes can be explicitly taken into account in solving soil heat transfer.

Some text was added in the section 2.1 and 2.3.

12. Line 148-153. I am confused with the reason to set these three cases. Would you please specify the purpose, which is to find a simple but effective enough coupling method? Is it not the most reliable method to include all physical processes?

**Response:** With these three cases, we can readily know the capacity of each soil physical parameterization. For instance, the effect of snowpack on underlying soil water and vapor transfer can not be realistically produced using the basic water and heat coupled model.

We can investigate the underlying physics and explain what happens after snowfall. The more complete the physical processes considered the model is more explainable. While under which condition the specific process is important and to what extent its importance is should be investigated. Here we would like to investigate the role of different soil physical processes in affecting the results. It can help to understand the underlying physics and to guide the future development of models.

13. Section 2.5. Please give some information about the snow distribution and observation in this site since the soil-snow interaction is the topic of this paper. Only soil information was given here.

**Response:** We added some text about the snow characteristics in this region (see Section 2.5). As the snow observation is scarce and difficult to collect in this site, we describe the snow distribution referring to the winter precipitation characteristics and other meteorological data. Line 208-209, 212-215.

14. Line 158. The station in TP, with shallow snowpack, I strongly suggest considering the solar radiation penetration.

**Response:** In UEB model, when the snowpack is shallow, the albedo is weighted between the snow albedo and bare ground albedo $A_{v/ir} = r\alpha_{g,v/ir} + (1-r)\alpha_{v/ir}$. The weighting factor $r = \left(1 - \frac{z}{h}\right)e^{-z/2h}$, in which the solar radiation penetration in the snow is exponentially attenuated as expressed by the exponential term. The incoming radiation is then altered and also the energy budget components.

We agree with your comments and the explicit consideration of the solar radiation penetration in the shallow snowpack and enriching the snowpack processes will be our next steps. See our revisions in Section 5.

15. RESULTS part. Where are the results on snow? Would you please give a quantitative evaluation of these results?

**Response:** We ran the simulation using the data from the Yakou super snow station in Qilian mountain. The comparisons of observed and STEMMUS-UEB simulated snow water equivalent dynamics were presented, together with the surface evaporation, soil moisture and temperature simulations. All the simulation results of the Yakou super snow station demonstrated the validity of STEMMUS-UEB model.

We added the site description in Section 2.5 and model results in Appendix B.

16. Line 319-321. It would be true in this region, but it would be evidently given here by using data, observation, references, and so on.

**Response:** We added the relevant case studies in Maqu (Li et al., 2017), which show that the snowpack is shallow and intermittent. On the other hand, it can be inferred from the less and temporal winter precipitation. See Section 4.1.

17. Figure 1. Please clearly present the coupling variables between UEB and STEMMUS-FT.

**Response:** Yes. The coupling variables were presented in the updated Figure 1. For the energy transfer, the variables are the ground heat conduction flux $Q_g$ and the convective heat flux due to the snowmelt $Q_m$. For the presented cases, soil surface temperature was set as the topsoil energy boundary condition. For the mass transfer, the snowmelt water flow $M_r$ is added on the soil surface as the coupling variable.

18. Figure 2. It is better to present the precipitation with columns but not lines. The color between precipitation and snow should be different. In addition, the improvement with the coupling snow module is not so noticeable.

**Response:** Yes. We replotted Figure 2 and presented the precipitation with columns.

For this simulation period, the precipitation/snowfall is intermittent and less. The total precipitation amount is only 6.1 mm. Nevertheless, the model with the snow module can well estimate the increase of surface albedo due to the presence of snowpack. The model without the snow module, however, presented no increase of surface albedo in response to the precipitation events.

The difference between models can be identified although it is not that noticeable as the snowfall is less and temporal.

19. Figures 3 and 4. The difference is tiny between two cases with and without snow modules.

**Response:** As the precipitation/snowfall is intermitted with less amount, the difference between the models with and without the snow module is indeed not that obvious. The difference between three cases with different soil physical processes is more significant here. It indicates that such amount of snowfall during this simulation period can only affect the hydrothermal regimes of the top sol layers and not enough to affect the deeper soil moisture and temperature dynamics.

On the other hand, we presented the results of soil moisture and temperature for Yakou station. The difference between the model with and without the snow module can be identified (Figure B4 and B5).

20. Figure 5. The improvement is not apparent from these results.

**Response:** The total precipitation during the simulation period is only 6.1 mm. The amount of snowfall and snow sublimation is thus not that much. Nevertheless, the difference between the model with and without snow module can be identified. The simulation of the cumulative LE was improved by the ACD and ACD-air models with the snow module. We added the additional plots in Figure 5, presenting the simulation bias (Simulation - Observation) for the model with and without the snow module, to make the improvement more visible.

To demonstrate the improvements using the model with and without snow module, we ran the simulations for the Yakou station (Figure B2 and B3). Surface evaporation was underestimated by the model without the snow module. Such underestimation was alleviated by the model with the snow module. Compared to the model without the snow module, the model with the snow module presented a closer correlation with the observed surface evaporation.

**Figures and Tables**

[Figure]

Figure R1. Scatter plot of the snow depth and albedo (Yakou station).

[Figure]

Figure R2. Time series of the snow depth, snow water equivalent (SWE), and albedo.

Three example periods were selected to illustrate the validity of using the indirect method, i.e., the albedo variation together with the ancillary meteorological data (air temperature Ta and precipitation), to identify the presence of snowpack and its lasting time (Table R1).

Table R1. The identification of snowpack using the direct evidence, i.e., the observed soil water equivalent (SWE, shaded with yellow color) and the indirect method, i.e., the albedo variation together with the ancillary meteorological data (air temperature Ta and precipitation) (shaded with blue color). The observed snow water equivalent is in 6-hour interval.

| TIME | Ta (°C) | Precipitation (mm) | Albedo | SWE (mm) | Remarks |
|---|---|---|---|---|---|
| 2016-10-10 12:30:00 | -0.8 | 0 | 0.14 | | |
| 2016-10-10 18:30:00 | -1.59 | 0 | 0.36 | | |
| 2016-10-11 00:30:00 | -5.24 | 0 | | | |
| 2016-10-11 06:30:00 | -6.73 | 0 | | 10.90 | |
| 2016-10-11 12:30:00 | -2.97 | 0.4 | 0.87 | 11.98 | |
| 2016-10-11 18:30:00 | -4.02 | 0 | 0.99 | 13.42 | First example period |
| 2016-10-12 00:30:00 | -4.44 | 0 | | 15.42 | |
| 2016-10-12 06:30:00 | -5.19 | 0 | | 16.74 | |
| 2016-10-12 12:30:00 | -3 | 0 | 0.62 | 17.22 | |
| 2016-10-12 18:30:00 | -1.45 | 0 | 0.61 | 17.17 | |
| 2016-10-13 00:30:00 | -2.84 | 0 | | 16.30 | |
| 2016-10-13 06:30:00 | -5.14 | 0 | | 15.61 | |
| 2016-10-13 12:30:00 | -0.37 | 0 | 0.18 | | |
| 2017-01-28 12:30:00 | -13.7 | 0 | 0.19 | | |
| 2017-01-28 18:30:00 | -14.32 | 0 | | | |
| 2017-01-29 00:30:00 | -17.1 | 0 | | | |
| 2017-01-29 06:30:00 | -15.15 | 0 | | 2.51 | |
| 2017-01-29 12:30:00 | -12.32 | 0 | 0.64 | 4.59 | |
| 2017-01-29 18:30:00 | -9.76 | 0 | | 6.69 | |
| 2017-01-30 00:30:00 | -11.82 | 0 | | 8.09 | |
| 2017-01-30 06:30:00 | -12.68 | 0 | | 9.26 | |
| 2017-01-30 12:30:00 | -8.95 | 0 | 0.61 | 8.69 | |
| 2017-01-30 18:30:00 | -9.58 | 0 | 0.95 | 8.31 | |
| 2017-01-31 00:30:00 | -11.71 | 0 | | 7.84 | |
| 2017-01-31 06:30:00 | -13.47 | 0 | | 7.01 | Second example period |
| 2017-01-31 12:30:00 | -10.24 | 0 | 0.51 | 7.18 | |
| 2017-01-31 18:30:00 | -9.76 | 0 | 0.85 | 6.40 | |
| 2017-02-01 00:30:00 | -11.95 | 0 | | 5.93 | |
| 2017-02-01 06:30:00 | -15.5 | 0 | | 4.75 | |
| 2017-02-01 12:30:00 | -10.63 | 0 | 0.41 | 4.71 | |
| 2017-02-01 18:30:00 | -8.66 | 0 | | 5.86 | |
| 2017-02-02 00:30:00 | -10.58 | 0 | | 6.12 | |
| 2017-02-02 06:30:00 | -12.15 | 0 | | 5.85 | |
| 2017-02-02 12:30:00 | -9.47 | 0 | 0.32 | 4.49 | |
| 2017-02-02 18:30:00 | -8.17 | 0 | | 3.82 | |
| 2017-02-03 00:30:00 | -10.22 | 0 | | 4.27 | |
| 2017-02-03 06:30:00 | -12.4 | 0 | | 4.27 | |

| Date/Time | | | | | |
|---|---|---|---|---|---|
| 2017-02-03 12:30:00 | -7.69 | 0 | | 0.29 | 4.00 |
| 2017-02-03 18:30:00 | -7.73 | 0 | | 0.23 | 4.06 |
| 2017-02-04 00:30:00 | -8.59 | 0 | | | 3.50 |
| 2017-02-04 06:30:00 | -8.37 | 0 | | | 2.08 |
| 2017-02-04 12:30:00 | -5.59 | 0 | | 0.23 | |
| 2017-02-06 12:30:00 | -2.91 | 0 | | 0.19 | |
| 2017-02-06 18:30:00 | -13.13 | 0 | | 0.49 | |
| 2017-02-07 00:30:00 | -17.7 | 0 | | | |
| 2017-02-07 06:30:00 | -19.04 | 0 | | | |
| 2017-02-07 12:30:00 | -16.09 | 0 | | 0.30 | 1.90 |
| 2017-02-07 18:30:00 | -17.33 | 0 | | 0.77 | 2.52 |
| 2017-02-08 00:30:00 | -18.17 | 0 | | | 3.54 |
| 2017-02-08 06:30:00 | -18.25 | 0 | | | 4.61 |
| 2017-02-08 12:30:00 | -13.95 | 0 | | | 5.73 |
| 2017-02-08 18:30:00 | -15.16 | 0 | | 0.99 | 7.19 |
| 2017-02-09 00:30:00 | -17.3 | 0 | | | 7.44 |
| 2017-02-09 06:30:00 | -17.53 | 0 | | | 7.74 |
| 2017-02-09 12:30:00 | -13.56 | 0 | | 0.65 | 7.91 |
| 2017-02-09 18:30:00 | -12.43 | 0 | | | 7.64 |
| 2017-02-10 00:30:00 | -16.64 | 0 | | | 8.12 |
| 2017-02-10 06:30:00 | -17.43 | 0 | | | 7.71 |
| 2017-02-10 12:30:00 | -16.36 | 0 | | 0.87 | 5.99 |
| 2017-02-10 18:30:00 | -14.38 | 0 | | | 7.58 |
| 2017-02-11 00:30:00 | -16.07 | 0 | | | 8.05 |
| 2017-02-11 06:30:00 | -16.7 | 0 | | | 8.54 |
| 2017-02-11 12:30:00 | -11.54 | 0 | | 0.61 | 8.92 |
| 2017-02-11 18:30:00 | -10.01 | 0 | | | 8.93 |
| 2017-02-12 00:30:00 | -13.76 | 0 | | | 8.27 |
| 2017-02-12 06:30:00 | -15.37 | 0 | | | 8.03 |
| 2017-02-12 12:30:00 | -9.63 | 0 | | 0.59 | 7.02 |
| 2017-02-12 18:30:00 | -7.45 | 0 | | 0.93 | 6.61 |
| 2017-02-13 00:30:00 | -9.27 | 0 | | | 6.37 |
| 2017-02-13 06:30:00 | -12.22 | 0 | | | 5.83 |
| 2017-02-13 12:30:00 | -7.75 | 0 | | 0.51 | 5.71 |
| 2017-02-13 18:30:00 | -9.31 | 0 | | 0.88 | 5.79 |
| 2017-02-14 00:30:00 | -11.14 | 0 | | | 5.61 |
| 2017-02-14 06:30:00 | -14.02 | 0 | | | 5.51 |
| 2017-02-14 12:30:00 | -8.78 | 0 | | 0.46 | 5.55 |
| 2017-02-14 18:30:00 | -7.36 | 0 | | | 4.80 |
| 2017-02-15 00:30:00 | -10.56 | 0 | | | 4.61 |
| 2017-02-15 06:30:00 | -12.26 | 0 | | | 4.52 |
| 2017-02-15 12:30:00 | -7.71 | 0 | | 0.37 | 3.99 |
| 2017-02-15 18:30:00 | -3.45 | 0 | | 0.76 | 3.81 |

Third example period

| | | | | |
|---|---|---|---|---|
| 2017-02-16 00:30:00 | -6.3 | 0 | | 2.03 |
| 2017-02-16 06:30:00 | -7.07 | 0 | | 2.62 |
| 2017-02-16 12:30:00 | -9.74 | 0 | 0.29 | 2.33 |
| 2017-02-16 18:30:00 | -10.48 | 0 | | 1.78 |
| 2017-02-17 00:30:00 | -10.66 | 0 | | 1.88 |
| 2017-02-17 06:30:00 | -10.74 | 0 | | 2.05 |
| 2017-02-17 12:30:00 | -7.3 | 0 | 0.25 | 2.12 |
| 2017-02-17 18:30:00 | -4.69 | 0 | 0.57 | |
| 2017-02-18 00:30:00 | -6.31 | 0 | | |
| 2017-02-18 06:30:00 | -8.64 | 0 | | 1.26 |
| 2017-02-18 12:30:00 | -5.53 | 0 | 0.21 | |

**Reference**

Che, T., Li, X., Liu, S., Li, H., Xu, Z., Tan, J., Zhang, Y., Ren, Z., Xiao, L., Deng, J., Jin, R., Ma, M., Wang, J., and Yang, X.: Integrated hydrometeorological, snow and frozen-ground observations in the alpine region of the Heihe River Basin, China, Earth System Science Data, 11, 1483-1499, https://doi.org/10.5194/essd-11-1483-2019, 2019.

Li, H., Li, X., Yang, D., Wang, J., Gao, B., Pan, X., Zhang, Y., and Hao, X.: Tracing Snowmelt Paths in an Integrated Hydrological Model for Understanding Seasonal Snowmelt Contribution at Basin Scale, Journal of Geophysical Research: Atmospheres, 124, 8874-8895, https://doi.org/10.1029/2019JD030760, 2019.

Li, D., Wen, L., Long, X., and Chen, S.: Observation Study on Effects of Snow Cover on Local Micro Meteorological Characteristics in Maqu, Plateau Meteorology, 36, 330-339, https://doi.org/10.7522/j.issn.1000-0534.2016.00074, 2017.

Prunty, L., and Bell, J.: Infiltration rate vs. gas composition and pressure in soil columns, Soil Sci Soc Am J, 71, 1473-1475, https://doi.org/10.2136/sssaj2007.0072N, 2007.

Touma, J., and Vauclin, M.: Experimental and numerical analysis of two-phase infiltration in a partially saturated soil, Transport in Porous Media, 1, 27-55, https://doi.org/10.1007/BF01036524, 1986.

Wicky, J., and Hauck, C.: Numerical modelling of convective heat transport by air flow in permafrost talus slopes, Cryosphere, 11, 1311-1325, https://doi.org/10.5194/tc-11-1311-2017, 2017.

Zeng, Y., Su, Z., Wan, L., and Wen, J.: A simulation analysis of the advective effect on evaporation using a two-phase heat and mass flow model, Water Resour Res, 47, W10529, https://doi.org/10.1029/2011WR010701, 2011a.

Zeng, Y., Su, Z., Wan, L., and Wen, J.: Numerical analysis of air-water-heat flow in unsaturated soil: Is it necessary to consider airflow in land surface models?, Journal of Geophysical Research: Atmospheres, 116, D20107, https://doi.org/10.1029/2011JD015835, 2011b.

Zeng, Y. J., and Su, Z. B.: STEMMUS : Simultaneous Transfer of Engery, Mass and Momentum in Unsaturated Soil, ISBN: 978-90-6164-351-7, University of Twente, Faculty of Geo-Information and Earth Observation (ITC), Enschede, 2013.